# A common *MET* polymorphism harnesses HER2 signaling to drive aggressive squamous cell carcinoma

Li Ren Kong [1,2✉], Nur Afiqah Binte Mohamed Salleh[1], Richard Weijie Ong[3], Tuan Zea Tan [1], Nicholas L. Syn [1,4], Robby Miguel Goh[4], Chee Wai Fhu[1], Daniel S.W. Tan[5,6,7], N. Gopalakrishna Iyer[1,8,9], Srinivasaraghavan Kannan [10], Chandra S. Verma[10,11,12], Yaw Chyn Lim[13], Ross Soo[4], Jingshan Ho[4], Yiqing Huang[4], Joline S.J. Lim[1,4], Benedict Junrong Yan[13], Min En Nga[13], Seng Gee Lim[14,15], H. Phillip Koeffler[1,4,14,16], Soo Chin Lee[1,4], Dennis Kappei [1], Huynh The Hung[3] & Boon Cher Goh [1,4,17✉]

c-MET receptors are activated in cancers through genomic events like tyrosine kinase domain mutations, juxtamembrane splicing mutation and amplified copy numbers, which can be inhibited by c-MET small molecule inhibitors. Here, we discover that the most common polymorphism known to affect *MET* gene (N375S), involving the semaphorin domain, confers exquisite binding affinity for HER2 and enables MET^N375S to interact with HER2 in a ligand-independent fashion. The resultant MET^N375S/HER2 dimer transduces potent proliferative, pro-invasive and pro-metastatic cues through the HER2 signaling axis to drive aggressive squamous cell carcinomas of the head and neck (HNSCC) and lung (LUSC), and is associated with poor prognosis. Accordingly, HER2 blockers, but not c-MET inhibitors, are paradoxically effective at restraining in vivo and in vitro models expressing MET^N375S. These results establish MET^N375S as a biologically distinct and clinically actionable molecular subset of SCCs that are uniquely amenable to HER2 blocking therapies.

[1] Cancer Science Institute of Singapore, National University of Singapore, Singapore 117599, Singapore. [2] Medical Research Council Cancer Unit, University of Cambridge, Cambridge CB2 0XZ, UK. [3] Laboratory of Molecular Endocrinology, National Cancer Centre Singapore, Singapore 169610, Singapore. [4] Department of Hematology-Oncology, National University Cancer Institute, Singapore 119074, Singapore. [5] Cancer Therapeutics Research Laboratory, National Cancer Centre Singapore, Singapore 169610, Singapore. [6] Division of Medical Oncology, National Cancer Centre Singapore, Singapore 169610, Singapore. [7] Genome Institute of Singapore, Agency for Science, Technology & Research (A*STAR), Singapore 138672, Singapore. [8] Office of Clinical Sciences, Duke-NUS Medical School, Singapore 169857, Singapore. [9] Division of Surgical Oncology, National Cancer Centre Singapore, Singapore 169610, Singapore. [10] Bioinformatics Institute, Agency for Science, Technology, and Research (A*STAR), Singapore 138671, Singapore. [11] Department of Biological Sciences, National University of Singapore, Singapore 119228, Singapore. [12] School of Biological Sciences, Nanyang Technological University, Singapore 639798, Singapore. [13] Department of Pathology, National University Health System, Singapore 119074, Singapore. [14] Department of Medicine, Yong Loo Lin School of Medicine, National University of Singapore, Singapore 119228, Singapore. [15] Division of Gastroenterology and Hepatology, National University Health System, Singapore 119074, Singapore. [16] Department of Hematology and Oncology, Cedars-Sinai Medical Center, Los Angeles, CA 90048, USA. [17] Department of Pharmacology, Yong Loo Lin School of Medicine, National University of Singapore, Singapore 119228, Singapore. ✉email: csiklr@nus.edu.sg; phcgbc@nus.edu.sg

The MET receptor tyrosine kinase (RTK) is essential for embryological and organ development[1,2]. Physiological activation of MET through binding of its only known ligand, hepatocyte growth factor (HGF)[3] induces receptor oligomerization, autophosphorylation, and effector kinase activation—involving pathways such as GRB2-RAS-MAPK, PI3K-Akt-mTOR[4], Src[4], and STAT3[5]—which orchestrate cellular motility[6], invasion[7], proliferation[8], angiogenesis[9], and differentiation[10]. Many human carcinomas have been found to constitutively activate MET via protein overexpression or gene amplification, which is associated with poorer survival[11,12] and resistance to EGFR tyrosine kinase inhibitors (TKIs) in EGFR mutant lung adenocarcinoma[13]. More recently, exon 14 skipping mutations in the juxtamembrane domain have emerged as an example of a therapeutically actionable biomarker conferring sensitivity to MET inhibitors[14,15].

The extracellular domain of MET consists of the Sema, PSI (plexin-semaphorin-integrin), and four IPT (Ig-like, plexins, transcription factors) subdomains. The Sema domain plays an essential function in ligand binding and receptor dimerization[16], but its role in cancer cells is not well delineated. A germinal MET polymorphism, Asn375Ser (N375S) residing in the Sema domain, has been found in ~10% of individuals of east and south Asian descent[17]. To our knowledge, the MET$^{N375S}$ polymorphism has not been definitively shown to increase cancer susceptibility, despite causing conformational changes at the ligand-binding site[18]. However, the lack of a clear association with cancer risk appears to belie the true pathogenic potential of MET$^{N375S}$, as we demonstrate in this study that the oncogenic effects of MET$^{N375S}$ are primarily manifested only in patients with active malignancies. In this study, we characterize the biologically- and clinically aggressive phenotype driven by MET$^{N375S}$ in LUSC and HNSCC, elucidate the intriguing mechanism by which MET$^{N375S}$ co-opts HER2 signaling to drive SCCs, and crucially, translate our findings into therapeutically cogent interventions with the successful therapy of tumor-bearing animals using commercially-available HER2 inhibitors. Our results therefore provide a strong clinical foundation for treating MET$^{N375S}$ SCC patients with HER2-targeted therapies.

## Results

**MET$^{N375S}$ is prognostic in LUSC and HNSCC.** We genotyped the MET$^{N375S}$ variant in germline DNA (gDNA) of several cohorts of patients (n = 1076) diagnosed with common cancers in the East Asian population in relation to healthy cancer-free controls, and determined the prevalence of the N375S polymorphism to be comparable in cancer patients relative to healthy, cancer-free controls (Fig. 1a, b). Patients included in this study were identified from the Pharmacogenetics Database of the National University Cancer Institute, Singapore, with gDNA samples and sufficient clinical follow-up data. Next, cancer patients treated with curative intent, either with surgery or chemoradiotherapy with or without adjuvant therapy, were stratified with respect to MET$^{N375S}$, and relapse-free survival (RFS) was studied as an indicator of the biological aggressiveness of cancers in MET$^{N375S}$-positive patients (Fig. 1c–j). For squamous cell head and neck cancer (HNSCC; n = 98), stage III and IV locally advanced disease of the buccal cavity, oropharynx, larynx, and hypopharynx who had undergone definitive treatment with combined modality therapy of either concurrent chemotherapy with weekly cisplatin (40 mg/m$^2$) and standard fractionation radiotherapy (66 Gy) or surgery with postoperative concurrent chemoradiation were included in the analysis so that stage and treatment-related prognoses would be relatively homogenous. The lung squamous cell carcinoma (LUSC; n = 123) cohort

consisted of patients with stage I to III disease who underwent surgery without adjuvant chemotherapy (before adjuvant chemotherapy became standard-of-care). The other cancer cohorts similarly comprised patients who have localized disease for which surgery was performed, and adjuvant treatment was administered according to standard practice. The RFS was significantly shorter in patients with the MET$^{N375S}$ polymorphism in both the SCC cohorts (Fig. 1c, d), suggesting a more aggressive oncogenic phenotype associated with this MET variant.

To confirm that the poor prognosis in these SCC cohorts were attributable to MET$^{N375S}$ polymorphism, amplicon-enriched next-generation sequencing was performed on 45 archival FFPE LUSC tissues that were retrieved from the Department of Pathology, National University of Singapore. We have earlier reported on the lack of driver oncogenes in these cases that include EGFR, ERBB2, KRAS, and PIK3CA genes[19]. Missense mutations were detected in 12 cases with 1 stop-gain mutation (Supplementary Fig. 1A). While N375S was the most prevalent alteration in these samples (9/45, 20%) (Supplementary Fig. 1B), we did not observe additional somatic mutations on the MET gene in these tumors. Apart from two cases, N375S mutation (seven out of nine cases) did not co-exist with known driver alterations (Supplementary Fig. 1C), further affirming the association of this MET variant with aggressive cancer phenotype.

**MET$^{N375S}$ promotes an aggressive tumor phenotype.** To characterize the phenotype associated with MET$^{N375S}$ in SCC, we generated isogenic cell lines expressing either wild-type or variant MET with turboGFP tag (tGFP) (MET$^{wt-tGFP}$ and MET$^{N375S-tGFP}$) in two LUSC lines (the epithelial H2170 cells and the p53-null mesenchymal Calu-1 cells) and two HNSCC lines (the cutaneous SCC13 cells and oral SCC UMSCC-1 cells). After single-colony selection, clones expressing comparable levels of MET$^{wt-tGFP}$ and MET$^{N375S-tGFP}$ were selected for subsequent functional studies. Introduction of MET$^{N375S}$ in LUSC cells significantly enhanced cell motility (Fig. 2a–d; Supplementary Fig. 2A, B), and anchorage-independent colony formation (Fig. 2e, f; Supplementary Fig. 2C, D). These oncogenic properties were attributable to the exogenous MET$^{N375S}$ protein, as silencing of MET was able to ablate the observed phenotypes (Fig. 2a–f). These findings were corroborated by in vivo subcutaneous H2170 xenograft models where the MET$^{N375S}$ variant tumors exhibited steeper growth gradients compared with their MET$^{wt}$ counterparts (Fig. 2g). In addition, while tail vein engraftment of both MET$^{wt-tGFP}$ and MET$^{N375S-tGFP}$ Calu-1 clones developed significant lung metastases compared with EV control (Fig. 2h), MET$^{N375S-tGFP}$ clones demonstrated enhanced metastatic potential by forming large "cannonball" metastatic nodules compared to MET$^{wt-tGFP}$ (Fig. 2h), with a greater tumor burden (Fig. 2i). These observations collectively demonstrate enhanced functional MET activity of the N375S variant, and are concordant with the shorter RFS observed in patients with HNSCC and LUSC harboring this variant.

To exclude the possibility that the observed phenotype could be attributed to the differential MET expression in these isogenic models, including transient overexpression, we derived LUSC cells with knock-in of MET$^{N375S}$ using CRISPR-Cas9 technology. We first showed that both MET$^{wt}$ and MET$^{N375S}$ were predominantly located at the membrane fractions (Supplementary Fig. 2E). Both H2170 and Calu-1 clones with homozygous N375S (MET$^{N375S/N375S}$) demonstrated spindle-shaped morphology (Supplementary Fig. 2F) and enhanced cellular motility (Supplementary Fig. 2G, H), which could be abrogated by MET silencing (Supplementary Fig. 2I, J).

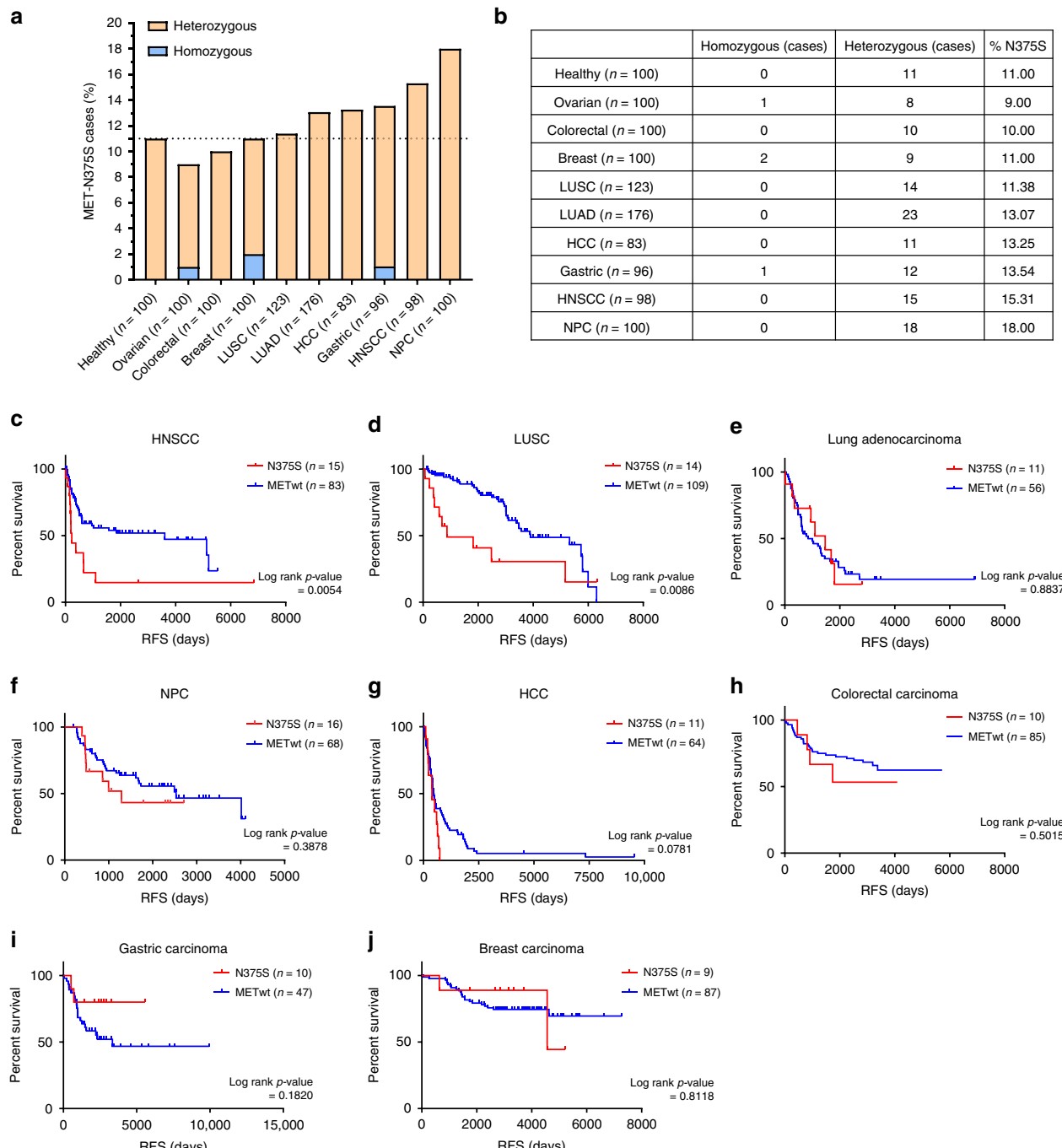

**Fig. 1 MET$^{N375S}$ mutation correlates with poorer relapse-free survival (RFS) in LUSC and HNSCC patients. a**, **b** Germline DNA from healthy volunteers and cancer patients, part of a Pharmacogenetics database with germline DNA derived from blood and with clinical follow-up data available, were sequenced with ddPCR using *MET* N375N (WT) and N375S-specific probes to determine the distribution and frequency of *MET (N375S)* genotype in Asian population. Graph (**a**) and table (**b**) showing the percentage and number of N375S + cases (heterozygous or homozygous) among healthy volunteers and cancer patients. **c**–**j** Relapse-free survival (RFS) of patients with locally advanced diseases who had undergone concurrent chemoradiotherapy or surgery were analyzed with Kaplan–Meier method and log-rank test. RFS (measured from time of treatment/surgery to relapse) for head and neck squamous cell carcinoma (**c**), lung squamous cell carcinoma (**d**), lung adenocarcinoma (**e**), nasopharyngeal carcinoma (**f**), hepatocellular carcinoma (**g**), colorectal carcinoma (**h**), gastric carcinoma (**i**), and breast carcinoma (**j**). Subjects who have not reached study-defined endpoint were censored (tick marks) from the analysis (Data cutoff point: January 2018).

**MET$^{N375S}$ signal transduction is different from MET$^{wt}$, and is not inhibited by MET inhibitors**. To compare the downstream signaling profiles between MET$^{N375S}$ and MET$^{wt}$, a human phospho-kinase antibody array analysis was conducted on the whole-cell lysates and showed that MET$^{N375S-tGFP}$ cells activated dissimilar signaling pathways compared to MET$^{wt-tGFP}$,

specifically the activation of Src (Fig. 3a–c). In concordance, higher expression of phosphorylated Src (Fig. 3d) and MET (Fig. 3d, e) were detected in MET$^{N375S}$ tumors.

To examine if the MET-driven signaling of N375S variant is a predictive biomarker for therapeutic intervention, H2170 isogenic cells were measured for viability after treatment with known MET

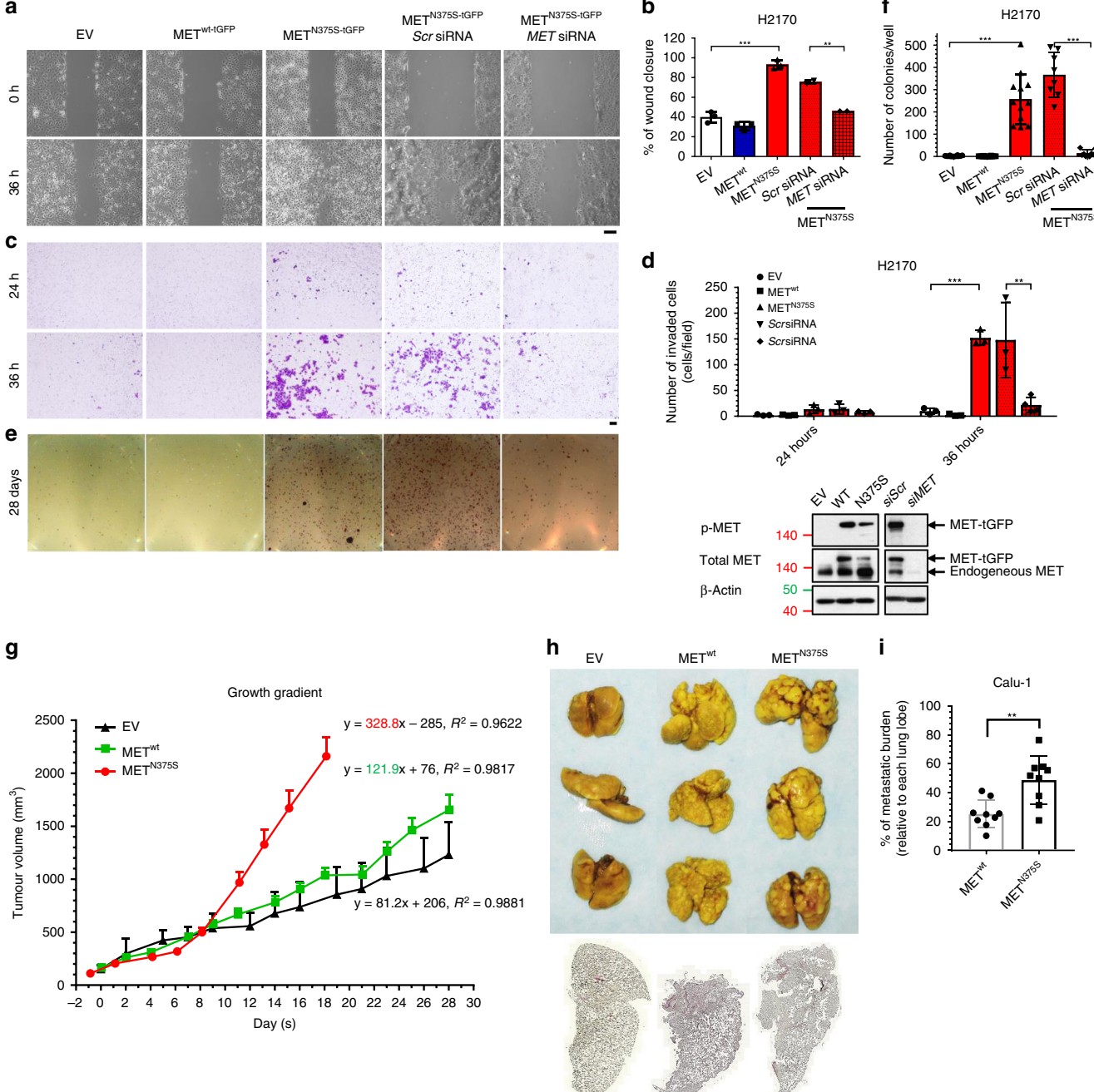

**Fig. 2 MET^N375S induces gain-of-function (GOF) phenotype in SCC cells. a–f** Cell migration, invasion, and colony-forming assays were performed on isogenic H2170 empty vector (EV), wild-type (MET^wt-tGFP), and N375S mutant (MET^N375S-tGFP) clones with and without siRNA silencing. In total, 10 nM of scrambled (*Scr*) or *MET* siRNA was used per transfection. Immunoblots demonstrating expression levels of the total and phosphorylated MET are shown (bottom right). β-Actin was used as a loading control. **a**, **b** Wounded areas (eight fields/sample) were imaged at 0 h and 36 h after monolayer cultures were scratched. **a** Representative images are shown. Scale bar: 200 μm. The wound-closure percentages for (**b**) EV, MET^wt-tGFP and MET^N375S-tGFP cells; *Scr* and *MET* siRNA transfected cells were quantified and expressed as mean ± SD ($n = 3$). Two-tailed Student's *t* test; *$P < 0.05$. **c**, **d** For invasion assay, cells seeded in Matrigel invasion chambers were fixed and stained at 24 or 36 h, respectively. **c** Representative images are shown. Scale bar: 200 μm. The number of cells in four random microscopic fields were quantified for EV, MET^wt-tGFP, and MET^N375S-tGFP cells; *Scr* and *MET* siRNA transfected cells (**d**), and expressed as mean ± SD in each field ($n = 3$). Two-tailed Student's *t* test; *$P < 0.05$. **e**, **f** Representative images of colony formation for each cell types, stained with MTT at assay endpoint (**e**). The number of colonies were quantified for EV, MET^wt-tGFP and MET^N375S-tGFP H2170 cells; *Scr* and *MET* siRNA transfected cells (**f**) and expressed as mean of triplicates ± SD ($n = 3$). Two-tailed Student's *t* test; *$P < 0.05$. **g** Tumor growth of EV, MET^wt-tGFP, and MET^N375S-tGFP cells in xenografts with growth gradient displayed on the chart, presented as mean growth ± SEM ($n = 5$). **h–i** Pulmonary metastases models were established with isogenic Calu-1 EV, MET^wt-tGFP, and MET^N375S-tGFP cells. **h** Macroscopic images of Bouin's-stained lungs from three individual female mice per group, dosed by tail vein injection of the respective cells. White spots on the lungs are visible metastatic colonies. Below, representative stitched scanned images of H&E-stained paraffin-embedded lung cross section. **i** Metastatic tumor burden was quantified based on percentage of tumor area in respective lung tissues, three lung lobes per mice, presented as mean ± SD ($n = 3$).

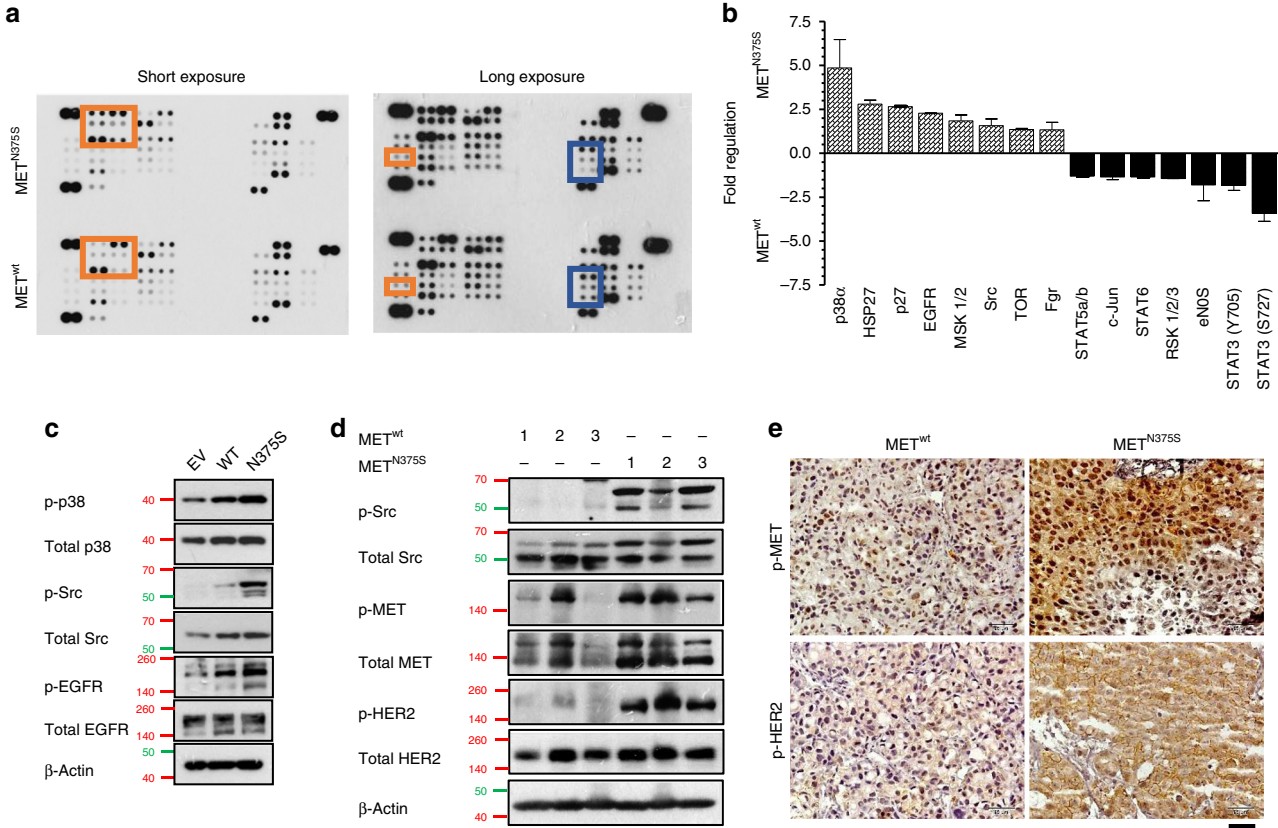

**Fig. 3 Aberrant activation of MAPK, Src, and HER2 signaling in carcinoma cells harboring MET^N375S mutation. a, b** Isogenic H2170 MET^wt-tGFP and MET^N375S-tGFP cells grown in optimal growth conditions were harvested and subjected to Proteome Profiler Human Phospho-Kinase Array. Dot plots with long and short chemiluminescence exposure are shown. Red indicates elevated phosphoproteins in MET^N375S; blue indicates downregulated proteins. Mean pixel density for each target spots was calculated with Quick Spots image analysis software, and tabulated as histogram demonstrating differential expressions of phosphoproteins in MET^wt-tGFP and MET^N375S-tGFP cells. Positive fold regulation indicates relative fold increase in MET^N375S-tGFP cells; negative fold regulation indicates relative fold increase in MET^wt-tGFP cells. Data expressed as mean of duplicate spots ± SD ($n = 1$). **c** Immunoblotting was performed to evaluate and validate the changes in the significantly regulated targets identified in (**a**). β-Actin was used as a loading control. **d, e** Xenograft tumors from Fig. 2g were harvested for phosphoprotein analyses. Immunoblots (**d**) and immunohistochemistry staining (**e**) showing the changes induced by ectopic MET^N375S expression in H2170 tumors. Representative blots and images are shown. Scale bar, 50 μm.

inhibitors cabozantinib, tepotinib, tivantinib, and crizotinib (Supplementary Fig. 3A–D). As shown, MET^N375S-tGFP cells did not exhibit differential half-maximal inhibitory concentration (IC50) values compared with MET^wt-tGFP cells, with the exception of tivantinib, which may have additional off-target effects[20]. Consistently, western blots of both MET^wt-tGFP and MET^N375S-tGFP cells showed comparable dephosphorylation of MET and Src after treatment with increasing doses of crizotinib and cabozantinib (Supplementary Fig. 3I). Moreover, both crizotinib and INC280—two tyrosine kinase inhibitors (TKIs) of MET—failed to elicit significant growth inhibition in MET^wt-tGFP and MET^N375S-tGFP tumors (Supplementary Fig. 3J–M), thereby recapitulating the in vitro sensitivity profiles. Importantly, the GOF of MET^N375S could not be abrogated by crizotinib (1 μM) (Supplementary Fig. S3O), despite complete suppression of MET phosphorylation (Supplementary Fig. S3I). These observations support earlier reports that MET^N375S is not responsive to MET inhibition[18,21]. We extended our analyses to inhibitors of EGFR (gefitinib), Src (saracatinib), PI3K (copanlisib), and ERK1/2 (trametinib), but none of these treatments induced significant growth inhibition of H2170 MET^N375S clones (Supplementary Fig. 3E–H), thereby suggesting the involvement of an upstream driver that contributes to the survival of MET^N375S-tGFP cells.

**MET^N375S preferentially binds HER2.** We postulated that the increased functional activity of MET^N375S may be related to either enhanced MET activity, or alternative pathway signaling through crosstalk with other growth factor receptors. To evaluate this, we first conducted comparative gene expression profiling on MET^wt-tGFP and MET^N375S-tGFP cells (Supplementary Fig. 4A), and confirmed there was neither *MET* amplification nor dysregulation of metastasis-inducing genes (such as fibronectin, RhoC, and thymosin β4)[22]. In addition, detailed analysis of the epithelial–mesenchymal transition (EMT) signature genes revealed identical EMT scores between the two MET variants (−0.59606 in MET^wt, −0.62169 in MET^N375S) (Supplementary Fig. 4B), indicating that the oncogenic signaling and phenotypic aggressiveness of MET^N375S occur independently of transcriptomic alterations.

Nuclear localization of MET is necessary for some of its functions[23,24]. To evaluate if the aggressive phenotype associated with MET^N375S is due to differential cellular localization, we first showed that both MET^wt and MET^N375S receptors are membrane bound (Supplementary Fig. 2E), further suggesting that the acquired oncogenic properties of the polymorphic variant are likely propagated through the plasma membrane. We hypothesized that the variant Sema domain may acquire conformational changes that alter receptor dynamics at the cell membrane, leading to aberrant cell signaling culminating in gain-of-function

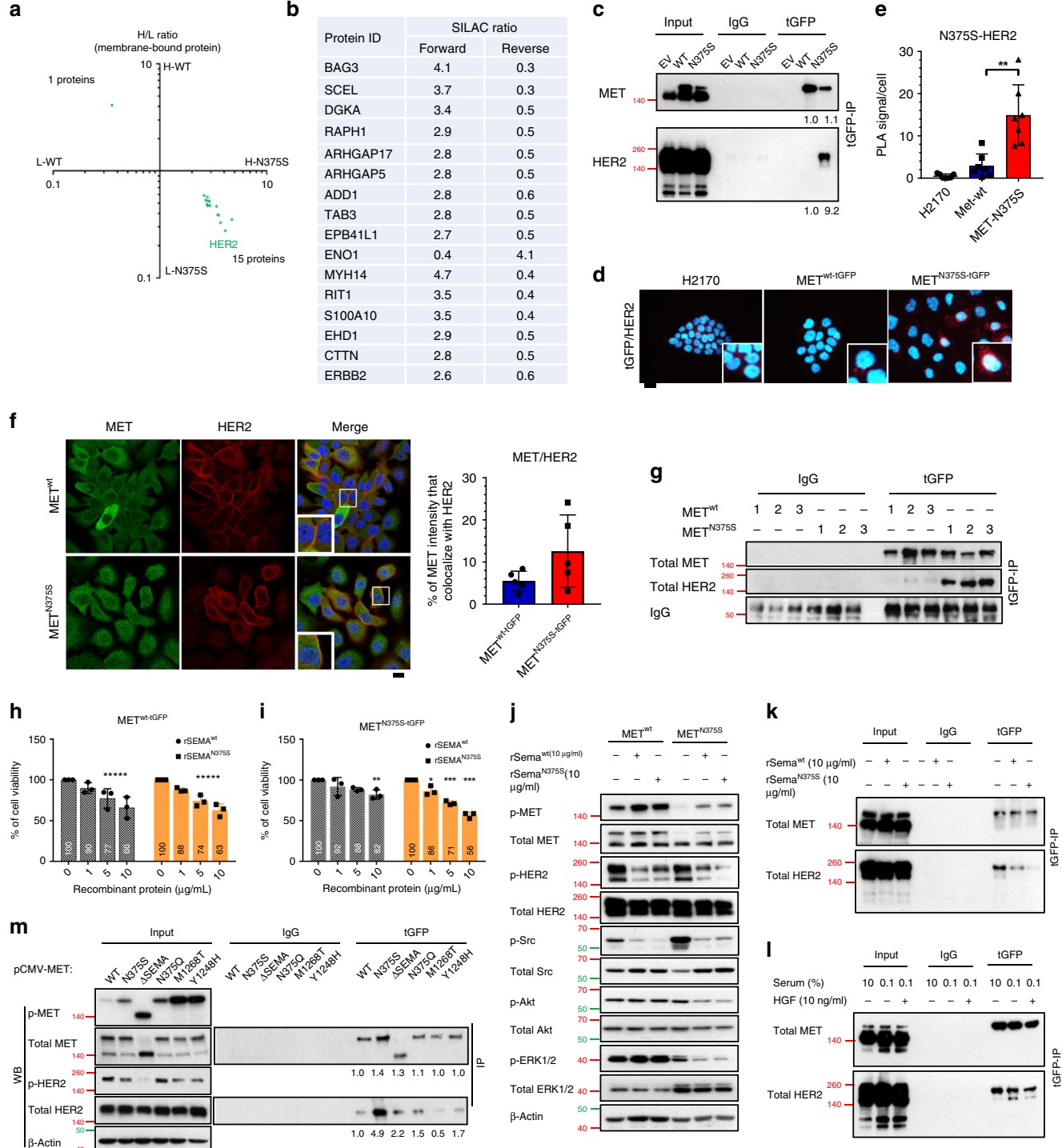

of MET. To elucidate this, we used SILAC-labeled protein extracts of isogenic MET$^{wt-tGFP}$ and MET$^{N375S-tGFP}$ cells to identify MET interacting partners. Specific binding to either the wild-type or MET variant is quantitatively detected by comparing cell lysates incubated with heavy ($^{15}$N- and $^{13}$C-labeled Lys and Arg) amino acids to those incubated with light, non-labeled amino acids using quantitative, high-resolution mass spectrometry after affinity capture. Here, background binders as well as common interaction partners between MET$^{wt}$ and MET$^{N375S}$ have 1:1 SILAC ratios, while specific binders to either variant display differential SILAC ratios after normalization with MET protein expressions. The top membrane-bound candidates with

normalized SILAC ratio above 2.5 (in both "forward" and "reverse" experiments where the labels were switched) were selected for validation (Fig. 4a, b); and among these, only HER2 strongly co-immunoprecipitated with exogenous MET$^{N375S}$ (Fig. 4c; Supplementary Fig. 5A). This preferential interaction was further substantiated with in situ proximity ligation assay (PLA) and fluorescent confocal imaging, that confirmed the proximal co-localization and possible heterodimerization of MET$^{N375S}$ and HER2 (Fig. 4d–f). We further demonstrated the specificity of this receptor dimerization by confirming the lack of receptor-binding preference of MET$^{N375S}$ to other TKs, such as EGFR and Src (Supplementary Fig. 5A);

**Fig. 4 The Sema domain of MET is critical for dimerization with HER2. a, b** Stable isotope labeling with amino acids in cell culture (SILAC) was performed to identify novel binding partners. MET$^{wt-tGFP}$ and MET$^{N375S-tGFP}$ cells were labeled with heavy (H) and light (L) amino acids. The cutoff values for SILAC ratios, after normalizing with MET expression, were set at (>2, <0.5) respectively. **a** Scatter plot of transformed MET$^{wt}$/MET$^{N375S}$ ratios of membrane-bound proteins. Both axes represent MET$^{wt}$(H)/MET$^{N375S}$(L) and MET$^{wt}$(L)/MET$^{N375S}$(H) ratios, respectively. **b** List of various membranous proteins identified in SILAC analysis found to be associated with MET$^{N375S}$. **c** Interaction of ectopic MET and endogenous HER2 in H2170 MET$^{wt-tGFP}$ and MET$^{N375S-tGFP}$ cells was detected with immunoprecipitation and immunoblotting. Left, input controls. Total MET and HER2 band intensities, normalized to input controls and relative to MET$^{wt}$, are shown below ($n = 5$). **d, e** Detection of MET/HER2 co-localization (red) in EV, MET$^{wt-tGFP}$, and MET$^{N375S-tGFP}$ cells with proximity ligation assay (PLA). Representative images are shown (**d**), with the PLA signals quantified (**e**) and expressed as the number of signals/cell ± SD ($n > 3$). Scale bar, 20 μm. **f** Representative confocal microscopy images of MET (Alexa-488; green) and HER2 (Alexa-594; red) in isogenic H2170 clones ($n = 2$). DAPI (blue) was used as nuclear counter stain. Co-localized proteins appeared in yellow. The smaller panels are detailed views of the outlined (white) squares in the respective images. Scale bar, 20 μm. The total MET fluorescence that co-localized with HER2 signal in each variant was tabulated on the right, data presented as mean ± SD (five fields in one representative experiment). **g** MET/HER2 interaction in H2170 MET$^{wt-tGFP}$ and MET$^{N375S-tGFP}$ tumors shown in Fig. 2g was detected with immunoprecipitation and immunoblotting. IgG was used as a loading control. **h–k** The role of Sema domain in MET$^{N375S-tGFP}$ cells was examined with recombinant Sema proteins, wild-type, rSema$^{wt}$; N375S mutant, rSema$^{N375S}$. Cell viability of MET$^{wt-tGFP}$ (**h**) and MET$^{N375S-tGFP}$ (**i**) cells after treatment with 1, 5, 10 μg/ml of rSema$^{wt}$ or rSema$^{N375S}$ for 72 h, presented as mean ± SD ($n = 3$). Two-tailed Student's $t$ test; \*$P < 0.05$, \*\*$P < 0.01$, \*\*\*$P < 0.001$. **j** Immunoblots showing the total and phosphorylated expressions of MET, HER2, Src, Akt, and ERK1/2 in lysates of the indicated cell lines after treatment with 10 μg/ml rSema. β-Actin was used as a loading control. **k** MET/HER2 interaction in H2170 MET$^{N375S-tGFP}$ cells was detected with immunoprecipitation and immunoblotting after treatment with 10 μg/ml rSema. Left, input controls. **l** MET/HER2 interaction in H2170 MET$^{N375S-tGFP}$ cells was detected after serum starvation (0.1% FBS), or co-incubated with HGF (0.1% FBS + 10 ng/ml HGF). Left, input controls. **m** HEK293 cells were transfected with 1 μg of either pCMV6-EV-tGFP vector, MET-wt, N375S, ΔSema, N375Q, M1268T, or Y1248H plasmid, together with pCMV6-ERBB2-DDK plasmid, for 24 h. MET/HER2 interaction in HEK293 cells was detected with immunoprecipitation and immunoblotting. Total MET and HER2 band intensities, relative to MET$^{wt}$, are shown below ($n = 3$). Left, input controls and phosphorylated proteins. β-Actin was used as a loading control.

whereas other prevalent MET variants of the Sema/juxtamembrane domains (E168D, S323G, R988C, and T1010I) did not preferentially co-immunoprecipitate with HER2 as compared to the N375S variant (Supplementary Fig. 5B).

As H2170 is a HER2-amplified (HER2$^+$) cell line, we further studied this binding between HER2 and MET$^{N375S}$ in HER2-non-amplified isogenic Calu-1 cells (Supplementary Fig. 6A–C), as well as in breast cancer cell lines that included SKBR3 (HER2-amplified) and MCF7 (HER2-non-amplified) (Supplementary Fig. 6D, E) to eliminate the possibility that the interaction is attributable to relative abundance of HER2 receptors. These substantiated the preferential interaction between MET$^{N375S}$ and HER2, as did similar assays in MET$^{wt-tGFP}$ and MET$^{N375S-tGFP}$ xenograft tumors (Fig. 4g). Collectively, the co-localization and strong binding affinity of both receptors in our assay models indicated that HER2 is a preferred interacting partner of MET$^{N375S}$.

**MET$^{N375S}$ receptor variant dimerizes with HER2.** The two-step activation of MET involves firstly the HGF ligand binding at the extracellular domain, followed by kinase activation at the intracellular domain. Given that Sema domain is necessary for ligand docking and MET dimerization[16], whereas the juxtamembrane domain regulates stability and nuclear trafficking of MET[25,26], we postulated that the interaction with HER2 requires an intact Sema domain. On the basis that Sema-domain decoy proteins could suppress RTK activities[16,27], we generated recombinant Sema proteins without the tyrosine kinase region—containing both the PSI domain and either WT (rSema$^{wt}$) or mutated (rSema$^{N375S}$) Sema sequences (Supplementary Table 1)—as inhibitor proteins to competitively disrupt the interaction of MET with its binding partners. While both rSema proteins dose-dependently reduced the cell viability of MET$^{wt-tGFP}$ cells (Fig. 4h), rSema$^{N375S}$ exhibited stronger growth inhibitory effects in MET$^{N375S-tGFP}$ cells compared with rSema$^{wt}$ (Fig. 4i). Mechanistically, treatment with either rSema induced transphosphorylation of MET but suppressed p-Src expression in both cell types (Fig. 4j), suggesting the occurrence of Sema–Sema interaction, but the lack of intracellular domain on rSema proteins impeded downstream signal transduction (Fig. 4j). Notably, rSema$^{N375S}$ profoundly repressed

HER2 phosphorylation in MET$^{N375S-tGFP}$ cells, together with the decrease of p-ERK1/2 and p-Akt (Fig. 4j). These were concordant with the more profound attenuation of HER2-binding to MET$^{N375S}$ by co-incubation with rSema$^{N375S}$ as compared to that with rSema$^{wt}$ (Fig. 4k), signifying the avid interaction in the presence of the N375S Sema-domain variant.

We next asked if a single substitution of asparagine at 375 by serine at the Sema domain is sufficient to foster the MET-HER2 interaction. It has been previously demonstrated that the conformational changes at MET$^{N375S}$ receptor reduced affinity to HGF[18], and we showed that HGF-depletion through serum starvation barely influenced the association of MET and HER2 (Fig. 4l), affirming ligand-independence for this interaction. We next transfected WT, N375S, ΔSEMA (Sema domain deleted), N375Q (amino acid substitution), M1268T (somatic activating), Y1248H (somatic activating)-mutant MET together with HER2 protein in HEK293 cells, and evaluated the receptor-binding preference to HER2. As compared with MET$^{wt}$, MET$^{N375S}$ variant profoundly co-immunoprecipitated with HER2, while Sema deletion (ΔSEMA) or modification (N375Q) substantially reduced this association (Fig. 4m, right). These observations strongly support an enhanced affinity for HER2 through Asn375Ser Sema modification, and explains the greater decoy effect of rSema$^{N375S}$ on cell viability and HER2 signaling in MET$^{N375S-tGFP}$ cells (Fig. 4i–k). Interestingly, oncogenic MET[28] (M1268T and Y1248H) displayed weakest binding to HER2 but possessed elevated MET phosphorylation (Fig. 4m, left), further establishing the distinctive oncogenic characteristics of N375S variant from other activating mutants. In concordance, xenograft MET$^{N375S-tGFP}$ tumors expressed substantial amounts of p-HER2 compared with MET$^{wt-tGFP}$ (Fig. 3d, e), indicating the strong dependence on HER2 in tumors with this particular MET variant.

**HER2 inhibitors are effective for MET$^{N375S}$-positive tumors.** Recent findings implicate the role of MET in the phosphorylation of PARP1 to impart insensitivity to PARP inhibitors through its kinase-dependent association to PARP1[24]; therefore, we examined the relevance of kinase activities on the association of MET$^{N375S}$ and HER2. Single or combination treatment of MET$^{N375S-tGFP}$ cells with crizotinib and lapatinib (HER2 TKI)

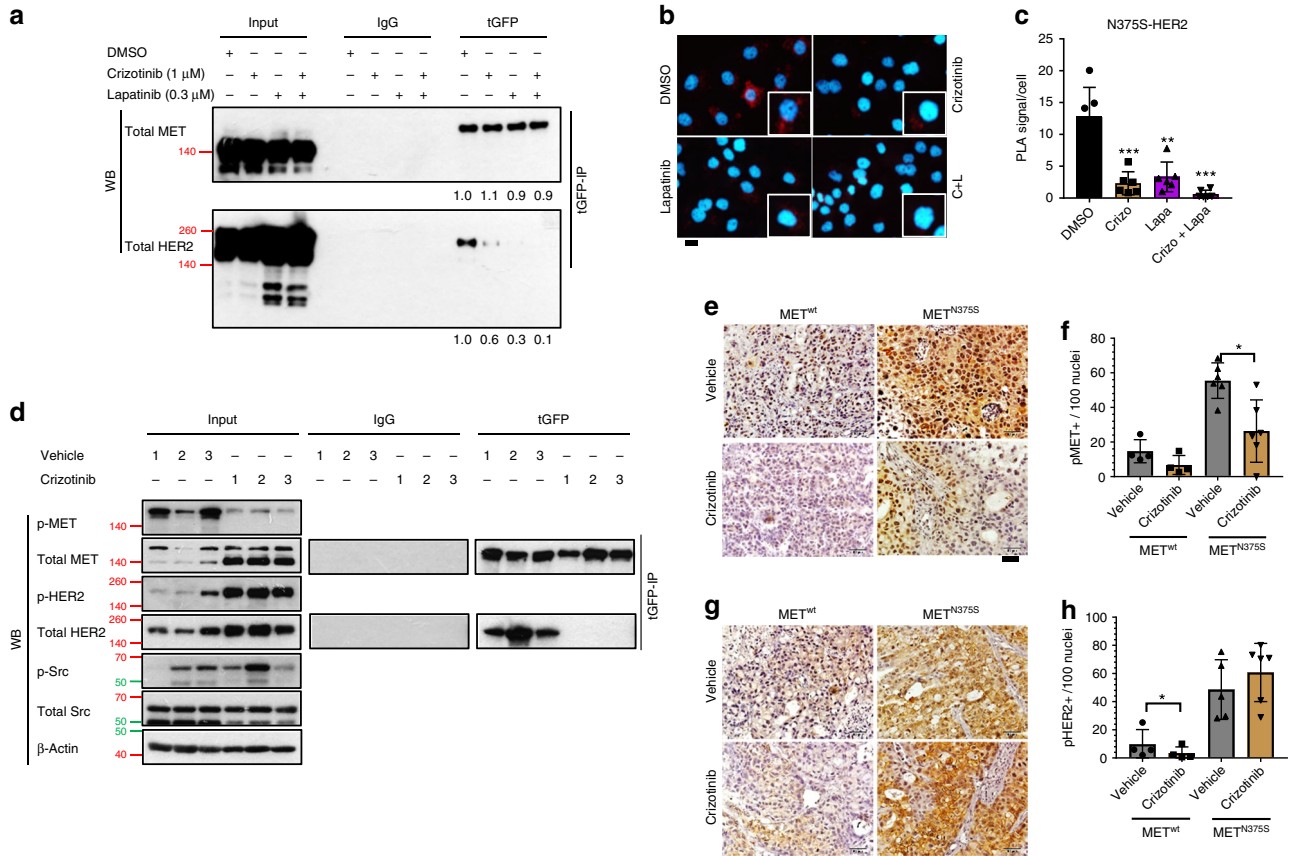

**Fig. 5 Interplay between MET and HER2 requires kinase activities. a–c** Isogenic H2170 MET$^{N375S-tGFP}$ cells were treated with vehicle (0.1% DMSO), MET inhibitor crizotinib (1 μM), HER2 inhibitor lapatinib (0.3 μM), or crizotinib/lapatinib combination. **a** Interaction of ectopic MET and endogenous HER2 was detected with immunoprecipitation and immunoblotting. Total MET and HER2 band intensities, relative to vehicle treated group, are shown below. Values represent average of three independent experiments. Left, input controls. **b** Detection of MET/HER2 co-localization (red) in MET$^{N375S-tGFP}$ cells with proximity ligation assay (PLA). Representative images are shown. Scale bar, 20 μm. **c** The PLA signals were quantified and expressed as number of signals/cell ± SD ($n > 3$). Crizo, crizotinib; Lapa, lapatinib. **d** MET/HER2 interaction in H2170 MET$^{N375S-tGFP}$ tumors from Supplementary Fig. 3J was detected with immunoprecipitation and immunoblotting after crizotinib treatment. Left, input controls. Cells were harvested 48 h after treatment, β-Actin was used as a loading control. **e–h** Immunohistochemistry staining of H2170 MET$^{wt-tGFP}$ and MET$^{N375S-tGFP}$ tumors from Supplementary Fig. 3J, K. Representative images for p-MET (**e**) and p-HER2 (**g**) staining are shown. Expression of p-MET (**f**) and p-HER2 (**h**) were quantified and expressed as mean of positive-staining/100 cells ± SD ($n = 5$). Scale bar, 50 μm. Two-tailed Student's $t$ test; *$P < 0.05$.

effectively impaired MET$^{N375S}$/HER2 interaction as confirmed by co-immunoprecipitation and PLA (Fig. 5a–c). In agreement with this observation, prolonged treatment of xenograft tumors with crizotinib abrogated MET phosphorylation and MET$^{N375S}$/HER2 interaction (Fig. 5d–f), which in turn led to a paradoxical increase in p-HER2 expression (Fig. 5d, g, h). This suggests that upon its activation by MET$^{N375S}$, HER2 remains hyperphosphorylated to continually promulgate oncogenic growth signals, further indicating that once activated by heterodimerization, HER2 remains constitutively active despite kinase inhibition of its partner receptor. This reliance on HER2 signaling would explain the lack of antitumor efficacy of MET inhibitors in xenograft models (Supplementary Fig. 3J–L).

Based on our observations, we postulated that targeting HER2 could be more effective in inhibiting the aggressive phenotype conferred by the variant MET. As shown, MET$^{N375S-tGFP}$ clones were more sensitive to lapatinib and afatinib (HER2 TKIs), with significantly lower IC$_{50}$ (Supplementary Fig. 7A, B), that was accompanied by substantial HER2 inhibition and dephosphorylation of ERK1/2 and Src (Supplementary Fig. 7C). We next compared the functional significance of MET and HER2 inhibition. As expected, *MET* silencing, but not crizotinib treatment, effectively hampered cell invasion in H2170

MET$^{N375S-tGFP}$ clones (Fig. 6a–c; Supplementary Fig. 8A). In contrast, HER2 silencing (*ERBB2* siRNA) and kinase inhibition (lapatinib, both singly and in combination with crizotinib) significantly neutralized cell invasion relative to scrambled siRNA or vehicle-treatment controls (Fig. 6a–c; Supplementary Fig. 8A). In addition, HER2 knockdown cells showed greater reduction of the invasion front (99% and 94%) as compared with MET knockdown cells (90% and 70%) in Calu-1 MET$^{N375S-tGFP}$ cells, as opposed to the observations in MET$^{wt-tGFP}$ cells (96% and 94% inhibition in si*MET* cells; 77% and 91% in si*ERBB2* cells) (Fig. 6d–f; Supplementary Fig. 8B). Consistently, silencing of MET or HER2 significantly attenuated anchorage-independent colony formation in H2170 MET$^{N375S-tGFP}$ cells (Fig. 6g), indicating that knockdown of either MET or HER2 could repress the oncogenic phenotype of the N375S variant. Collectively, these data suggest that MET$^{N375S-tGFP}$ cells attain the aggressive phenotype through interactions between intact MET and HER2 receptors, leading to HER2 phosphorylation that once activated is constitutively active and is irrepressible by MET kinase inhibition. Accordingly, therapeutic intervention of this oncogenic mechanism should focus on HER2 blockade in these tumors.

To determine if MET status influences tumor response to HER2 inhibition, we examined the colony formation and tumor

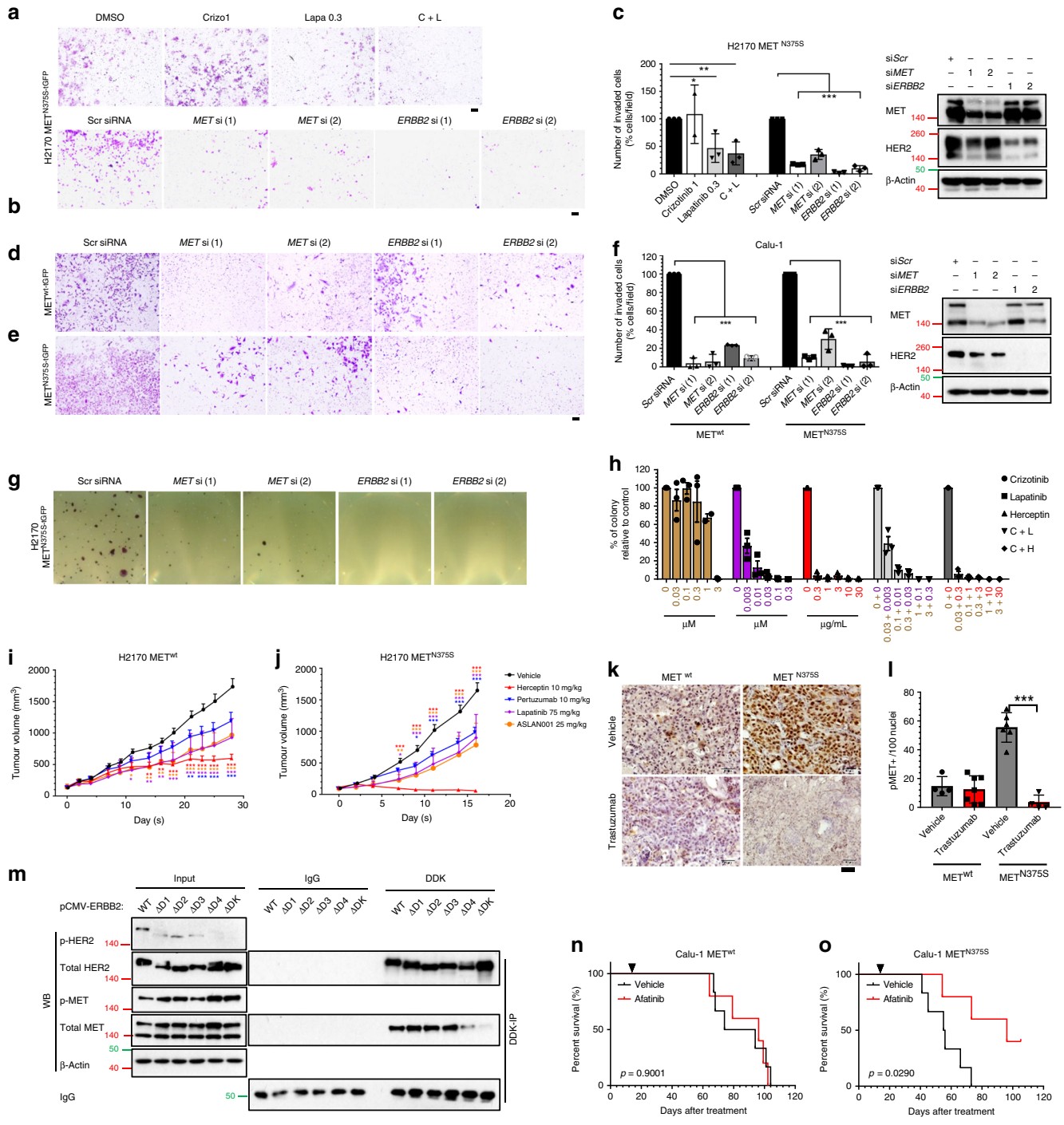

growth of isogenic H2170 clones treated with the HER2 TKIs (lapatinib, ASLAN001) and monoclonal antibodies (trastuzumab and pertuzumab). In contrast to crizotinib that achieved only modest effects at therapeutic doses[29], lapatinib and trastuzumab both potently inhibited anchorage-independent growth of MET[N375S-tGFP] cells (Fig. 6h; Supplementary Fig. 8C–E). Furthermore, combination treatment with crizotinib did not augment the growth inhibitory effect of lapatinib and trastuzumab (Fig. 6h; Supplementary Fig. 8F, G). Comparisons of in vivo efficacies between wild-type and polymorphic MET tumors demonstrated that clinically relevant concentrations of HER2 blockers, through suppression of p-HER2 (Supplementary Fig. 8J), exerted anti-proliferative effects in both H2170 cell types due to the HER2+

status (Fig. 6i, j; Supplementary Fig. 8H, I). Nonetheless, MET[N375S-tGFP] tumors were more sensitive, particularly to trastuzumab where tumor regression was attained at tolerable doses with significant dephosphorylation of MET in addition to the ablation of HER2 activity (Fig. 6k, l; Supplementary Fig. 8H, I), likely due to the interference with heterodimerization between HER2 and MET[N375S]. To evaluate the structural basis of HER2 dimerization with MET[N375S], HEK293 cells transfected with MET-N375S vector and either one of the HER2-mutant constructs (WT, ΔD1, ΔD2, ΔD3, ΔD4, or ΔTK) were studied with co-immunoprecipitation assays; intact HER2 subdomain IV was shown to be necessary for binding to MET[N375S], and that HER2 TK activation facilitates this interaction (Fig. 6m). This

**Fig. 6 Therapeutic targeting of MET$^{N375S}$ tumors using inhibitors of HER2 signaling. a–c** Cellular invasion on isogenic wild-type (MET$^{wt-tGFP}$) and N375S mutant (MET$^{N375S-tGFP}$) clones, treated with kinase inhibitors or siRNA silencing of MET or HER2, were being evaluated. **a** H2170 MET$^{N375S-tGFP}$ cells were treated with crizotinib (10 μM), lapatinib (0.3 μM) or a crizotinib/lapatinib combination for 36 h. Representative images of cell invasion are shown. **b** H2170 MET$^{N375S-tGFP}$ cells were co-incubated with 10 nM of the indicated siRNA overnight, and seeded in Matrigel invasion chambers for 36 h. Representative images are shown. Scale bar: 200 μm. **c** Percentage of invaded cells relative to vehicle control (0.1% DMSO) or *Scr* siRNA control were expressed as mean ± SD (*n* = 3). One-way ANOVA; *\*P* < 0.05, *\*\*P* < 0.01, *\*\*\*P* < 0.001. Left, immunoblots of siRNA-treated H2170 MET$^{N375S-tGFP}$ cells. For immunoblots, cells were harvested 48 h after siRNA treatment, and β-actin was used as a loading control. **d–f** Calu-1 MET$^{wt-tGFP}$ and MET$^{N375S-tGFP}$ cells were co-incubated with 10 nM of the indicated siRNA overnight, and seeded in Matrigel invasion chambers for 24 h. Representative images of cell invasion for MET$^{wt-tGFP}$ (**d**) and MET$^{N375S-tGFP}$ cells (**e**) are shown. Scale bar: 200 μm. **f** Percentage of invaded cells relative to *Scr* siRNA control in Calu-1 MET$^{wt-tGFP}$ and MET$^{N375S-tGFP}$ cells were expressed as mean ± SD (*n* = 3). Left, immunoblots of siRNA-treated Calu-1 MET$^{N375S-tGFP}$ cells. For immunoblots, cells were harvested 48 h after siRNA treatment, and β-actin was used as a loading control. One-way ANOVA; *\*P* < 0.05, *\*\*P* < 0.01, *\*\*\*P* < 0.001. **g, h** Anchorage-independent colony formation on isogenic H2170 MET$^{N375S-tGFP}$ cells, treated with kinase inhibitors or siRNA silencing of MET or HER2, were being evaluated. **g** H2170 MET$^{N375S-tGFP}$ cells were co-incubated with 10 nM of the indicated siRNA overnight, and seeded in soft agar for 4 weeks. Representative images are shown. **h** The number of colonies were quantified after treatment with crizotinib, lapatinib, trastuzumab, or in combination. Data were presented as percentage of colonies relative to vehicle control (0.1% DMSO) ± SD (*n* = 3). **i–l** Efficacies of HER2 inhibitors were evaluated in xenograft models. Tumor growth of MET$^{wt-tGFP}$ (**i**) and MET$^{N375S-tGFP}$ (**j**) xenografts after treatment with trastuzumab, pertuzumab, lapatinib, and ASLAN001 were expressed at mean ± SEM (*n* = 5). Two-way ANOVA; *\*P* < 0.05, *\*\*P* < 0.01, *\*\*\*P* < 0.001. **k** Immunohistochemistry staining showing the changes in p-MET after treatment with trastuzumab in MET$^{wt-tGFP}$ and MET$^{N375S-tGFP}$ tumors. Representative images are shown. Scale bar, 50 μm. **l** Expression of p-MET was quantified and expressed at mean of positive-staining/100 cells ± SD (*n* = 5). Two-tailed Student's *t* test; *\*P* < 0.05, *\*\*P* < 0.01, *\*\*\*P* < 0.001. **m** HEK293 cells were transfected with 1 μg of either pCMV6-ERBB2-wt, ΔD1, ΔD2, ΔD3, ΔD4, or ΔTK plasmid, together with pCMV6-MET-N375S plasmid, for 24 h. MET/HER2 interaction in HEK293 cells was detected with immunoprecipitation and immunoblotting. Left, input controls and phosphorylated proteins. β-Actin was used as a loading control. **n, o** Isogenic Calu-1 MET$^{wt-tGFP}$ (**n**) and MET$^{N375S-tGFP}$ (**o**) cells were engrafted into SCID mice, and treated daily with vehicle (*n* = 6) or 15 mg/kg afatinib (*n* = 5) starting 14 days after inoculation. Arrows indicate treatment start date. Kaplan–Meier analyses of the mice are shown.

also explains the sensitivity of MET$^{N375S-tGFP}$ xenografts to trastuzumab, which complexes with subdomain IV of HER2[30].

Using a primary culture derived from a HER2-non-amplified *MET* (N375S) heterozygous tongue SCC tumor (NCC-NPC7), we confirmed the increased susceptibility to HER2 inhibitors (lapatinib and afatinib), but not to crizotinib as compared to a HER2-non-amplified MET wild-type HNSCC cell line (SCC13) (Supplementary Fig. 9A). Mechanistically, crizotinib attenuated p-MET and p-Src while not affecting other kinases in NPC7 cells (Supplementary Fig. 9B). On the contrary, treatment with lapatinib and afatinib effectively abrogated kinase activities of HER2, EGFR, Akt, and ERK1/2 (Supplementary Fig. 9B). In addition, we confirmed that MET$^{N375S}$ variant in NCC-NPC7 cells co-immunoprecipitated with HER2, and similar to that in isogenic H2170 cells (Fig. 5a), this interaction could be abolished by kinase inhibition through crizotinib or/and lapatinib treatment (Supplementary Fig. 9C). Importantly, afatinib given at 15 mg/kg daily was found to significantly improve overall survival (log-rank test; *P* < 0.05) over the vehicle control in immunocompromised mice engrafted with MET$^{N375S-tGFP}$ cells (Fig. 6o). In contrast, mice engrafted with MET$^{wt-tGFP}$ cells survived longer and did not benefit from afatinib treatment (Fig. 6n), thereby confirming the aggressiveness of N375S tumors as well as the therapeutic potential of pharmacological HER2 inhibition in these tumors.

**Clinical correlation of MET$^{N375S}$ and HER2 in SCC.** To evaluate the clinical correlation between MET$^{N375S}$ and HER2, we conducted an immunohistochemistry study to understand their phosphorylation status in relation to MET genotype. Archival FFPE tissue microarray comprised normal lung tissues or LUSC tissues were retrieved. The expression of p-HER2 and p-MET were scored accordingly for the 45 LUSC tumor specimens with *MET* genotyped and sequenced (Fig. 1d; Supplementary Fig. 1). We showed that both HER2 and MET were not activated in normal lung tissues (Supplementary Fig. 10A), whereas 5 of the 36 MET$^{wt}$ tumors were found to be positive for p-HER2 (score ≥ 1+) with no p-MET expression (Supplementary Fig. 10B, Supplementary Table 2). Concordant with our in vivo observations,

the MET$^{N375S}$-positive tumors were demonstrated to have strong p-HER2 (9/9 with score ≥ 2+) and p-MET (8/9 with score ≥ 2+) (Supplementary Fig. 10C–E, Supplementary Table 2). The expressions of p-HER2 and p-MET were shown to be positively correlated (Fisher's exact test, *P* < 0.0001) (Supplementary Fig. 10F), indicative of increased HER2 activation in tumors with activated MET$^{N375S}$, and these MET$^{N375S}$-positive patients had significantly shorter RFS compared to MET$^{wt}$ patients (Supplementary Fig. 10G). This is in concordance with our postulation that MET$^{N375S}$ phosphorylation is essential for HER2 activation that promotes the aggressiveness of SCC. Indeed, HER2 blockade with afatinib, lapatinib, or trastuzumab did not demonstrate any antitumor efficacy in two MET$^{N375S}$-positive patient-derived xenografts (PDX) of hepatocellular carcinoma (HCC) (Supplementary Fig. 9D–G), in which phosphorylations of MET and HER2 were not prominent in the tumors (Supplementary Fig. 9H, I). This likely explains the tissue- and context-specificity of MET$^{N375S}$ on LUSC and HNSCC.

## Discussion
The oncogenic role of MET has been established in multiple cancers, with gene amplification, somatic mutations, and splicing variants being the driving mechanisms for oncogenic MET function, while rare germline-activating mutations in the tyrosine kinase (TK) domain have been reported in hereditary papillary renal cell carcinoma[28]. However, these events are rare in SCC. We describe here an aberrant oncogenic activity of a polymorphic Sema variant of MET, which when present, enhances the aggressiveness of SCC tumors in vitro and in vivo. Patients with HNSCC and LUSC who carry the MET$^{N375S}$ polymorphism have higher risk of disease recurrence, mediated by a novel mechanism of oncogenicity that involves enhanced dimerization to another oncogenic membrane-bound RTK receptor HER2 without HGF ligand activation (Supplementary Fig. 11). This mechanism of Sema-domain activation due to a single amino acid substitution adds to the other more established molecular mechanisms of oncogenic MET signaling, which mostly revolve around constitutive and persistent kinase activity.

It is interesting to note that Sema proteins have been shown to act as oncogenic ligands in the activation of RTKs that include MET and HER2[27]. Based on molecular modeling and simulations, we postulate that the asparagine-to-serine substitution could induce significant localized conformational changes (Supplementary Fig. 12A, B), which improve surface interactions between MET$^{N375S}$ and HER2 (Supplementary Fig. 12C, D), leading to gain-of-function of the polymorphic variant. Taken together with the imaging and biochemical characterization, our data collectively implicate that the N375S modification may reconfigure the Sema domain for MET interaction with HER2. The resulting MET$^{N375S}$ phosphorylation is necessary for HER2 activation, as rSema$^{N375S}$ proteins lacking the tyrosine kinase domain could inhibit HER2 phosphorylation. HER2 is rarely amplified or mutated in HNSCC/LUSC[31], but this activation through polymorphic MET may lead to constitutive HER2 signaling and drive these cancers. Our data further demonstrates that the heterodimeric signal between MET$^{N375S}$ and HER2 is specifically dependent on an intact subdomain IV on HER2, which is the binding site of trastuzumab, explaining the enhanced sensitivity of MET$^{N375S}$ tumors to trastuzumab relative to HER2 TKIs. While MET has been previously demonstrated to be a heterodimeric partner of HER2[32], our work on SCC shows a strong addiction of MET$^{N375S}$ tumors to HER2 signaling.

The HER2 receptor lacks a ligand-binding domain, and its activation relies heavily on heterodimerization with other ligand activated receptors, particularly HER3[30,33]. Notwithstanding, HER2 mediates potent downstream signaling when activated through its highly catalytic TK domain[34], and is a strong inducer of cell motility and metastasis[35]. Our findings strongly indicate that phosphorylated MET$^{N375S}$ leads to constitutively active HER2 that mediates signaling regardless of MET inhibition. This insensitivity to MET inhibition is surprising and remains unexplained; possible reasons could include involvement of other HER2 activating mechanisms. Conversely, HER2 inhibitors effectively suppress oncogenic MET$^{N375S}$ signaling in SCC tumors, as evidenced by their strong anti-proliferative activity both in vitro and in vivo. This adds on to the growing recognition that HER2 inhibition could be effective in TKI-relapsed MET-driven tumors[36], and MET$^{N375S}$ exemplifies emerging evidence that germline polymorphisms can drive malignancy and represent bona fide biomarkers to select therapeutic agents clinically. Although HER2-amplified H2170 was primarily used in the study which may influence sensitivity to HER2 inhibitors, we recapitulated the experimental findings in non-HER2 overexpressing cells like Calu-1 and patient-derived NPC7 cells.

The treatment of SCC is often challenging due to the lack of common actionable driver oncogenes. Cetuximab plus platinum-based chemotherapy has been established as first-line treatment in recurrent or metastatic HNSCC as EGFR is often hyperexpressed in this disease[37,38], whereas standard-of-care regimens comprise of platinum-based doublet in combination with vinorelbine, gemcitabine or taxane for LUSC, and has remained largely unchanged over the past decades[39]. To address this substantial unmet therapeutic need, we are currently conducting a clinical trial to treat patients with metastatic HNSCC and LUSC who screened positive for the MET$^{N375S}$ polymorphism with HER2 blockade (NCT03938012). However, it is unclear at this point which anti-HER2 therapy would be optimal clinically; based on our in vivo models, trastuzumab appears to be the most promising.

Currently, the oncogenic role of the N375S variant in other cancers remains unclear. Theoretically, tumors overexpressing MET should be subjected to the similar aggressive influences of the polymorphic receptor, notwithstanding, our data demonstrated the ineffectiveness of HER2 inhibition in MET$^{N375S}$-

positive HCC, suggesting that this strategy is tumor-context dependent, and is effective particularly in tumors with activated MET$^{N375S}$. Our data do not suggest that the MET$^{N375S}$ polymorphism increases cancer susceptibility; however, whether the risks are elevated in the homozygous state needs further clarification, as the majority of the individuals in our study were heterozygotes. In the clinical context, HER2 amplification is common in a variety of cancers— such as breast and hepatocellular carcinoma—and how this potentially cooperates with MET$^{N375S}$ to promote malignancy is unclear, though our experimental models have shown that HER2 amplification is not essential for the oncogenic phenotype conferred by the MET variant. Nonetheless, the lack of efficacy of HER2 inhibitors in MET$^{N375S}$ HCC tumors with low p-HER2 expression suggest that other mechanisms are more dominant in HCC, and that HER2 activation is necessary for therapeutic impact. Our current work contributes to the emerging notion that germline genetic events could impact the selection of oncotherapeutic agents, and in this case, specifically in SCC. While the *MET* (N375S) polymorphism may be more prevalent among Asians, the findings should be broadly applicable to all patients selected on the basis of this biomarker.

## Methods

**Chemicals, antibodies, and recombinant proteins**. For inhibitor studies, crizotinib (#S1068), INC280 (#2788), gefitinib (#S1025), saracatinib (#S1066), copanlisib (#S2802), trametinib (#S2673), lapatinib (#S2111), and afatinib (#S1011) were purchased from Selleckchem. All compounds were dissolved in Dimethyl sulfoxide (DMSO, Sigma Aldrich, #D8418). The monoclonal recombinant antibody trastuzumab (A2007) and pertuzumab (A2008) were also obtained from Selleckchem. ASLAN001 was provided by Huynh Hung as a gift.

For immunoblotting, p-MET (Tyr1234/1235, #3077), total MET (#8198), p-HER2 (Tyr1221/1222, #2243), total HER2 (#2165), p-p38 (Thr180/Tyr182, #4511), total p38 (#8690), p-Src (Tyr416, #2101), total Src (#2108), p-ERK1/2 (Thr202/Tyr204, #4377), total ERK1/2 (#4695), p-mTOR (Ser2448, #2971), total mTOR (#2172), p-Stat3 (Tyr705, #9131), total Stat3 (#4904), p-EGFR (Tyr1068, #2234), total EGFR (#2646), p-Akt (Ser473, #4060), total Akt (#9272), α-tubulin (#2125), horseradish peroxidase (HRP)-conjugated MET (#24294), HER2 (#60388), HSP90 (#4874), and β-actin (#5125) antibodies were purchased from Cell Signaling Technology; turboGFP antibody was from Origene (#TA150041); pan-Cadherin antibody (#ab22744) from Abcam. Anti-mouse (#7076) and -rabbit (#7074) HRP-conjugated secondary antibodies were obtained from Cell Signaling Technologies. All antibodies were used at 1:2000 dilution. For co-immunoprecipitation, turboGFP antibody from Origene was used, together with mouse IgG (#5415) antibody from Cell Signaling Technology. For Duolink PLA assay, rabbit total HER2 was paired with either turboGFP or total MET (Cell Signaling Technology, #8741) of mouse origin, and used at 1:1000 dilution. For immunofluorescence staining, total HER2 and total MET (Cell Signaling Technology, #8741) were used at 1:1000 dilution; secondary goat anti-mouse Alexa-488 (Thermo Fisher, A-11001 and goat anti-rabbit Alexa-594 (Thermo Fisher, A-11012) were used at 1:1000 dilution. For immunohistochemistry staining, p-MET and p-HER2 antibodies were used at 1:50 dilution.

Recombinant Sema proteins (rSema) were designed as described previously with modifications[40] (WT and N375S) (Supplementary Table 1), and generated by MyBioSource. The rSema contains both the Sema and PSI domains, and synthesized with a His tag using pET30a vector. Proteins were eluted with a buffer containing 10 mM Tris pH 8.0, 1 mM EDTA, and 10% glycerol. Endotoxin removal was carried out (<1.0 EU per 1 μg of protein). Recombinant protein has purity of ≥ 85%, and a size of 63.4 kDa.

**Cell culture**. All cell lines were obtained directly from the American Type Culture Collection (ATCC), and were cultured in RPMI 1640 medium supplemented with 10% fetal bovine serum, 2 mM L-glutamine, 100 μg/mL streptomycin, and 100 U/mL penicillin in a humidified 37 °C incubator. For SILAC labeling, cells were incubated in RPMI 1640 (-Arg, -Lys) medium containing 10% dialyzed fetal bovine serum (Thermo) supplemented with 84 mg/l $^{13}C_6^{15}N_4$ L-arginine and 50 mg/l $^{13}C_6^{15}N_2$ L-lysine (Cambridge Isotope) or the corresponding non-labeled amino acids, respectively. Cell lines were authenticated by short tandem repeat (STR) profiling using Promega GenePrint 10 system (#B9510). Primary culture cell line (NPC7) was obtained from N GopalakrishnaIyer (National Cancer Centre, Singapore) as a gift. HNSCC cell lines (SCC13 and UMSCC-1) were provided by H. Phillip Koeffler, and cultured in the DMEM medium with standard supplements. Isogenic cell lines stably expressing pCMV6-AC-GFP vector, pCMV6-MET (wt), and pCMV6-MET (N375S) plasmids, respectively, were generated by transfection and single-cell clonal selection. Cell lines homozygous of MET$^{N375S/N375S}$ were established using CRISPR-

Cas9 genome-editing technique, as previously described[41]. Single-cell colonies were selected and validated using Sanger Sequencing and ddPCR.

**Plasmid transfection, mutagenesis, and RNA interference.** pCMV6-AC-turboGFP (tGFP) vector, pCMV6-MET-tGFP, and pCMV6-ERBB2-DDK plasmids were obtained from Origene (#PS100010, #RG217003, and #RC12583). Site-directed mutagenesis was performed on pCMV6-MET-tGFP for substitution of glutamate-to-aspartate at 168 (E168D), serine-to-glycine at 323 (S323G), asparagine-to-serine at 375 (MET-N375S), asparagine-to-glutamine at 375 (N375Q), arginine-to-cysteine at 988 (R988C), threonine-to-isoleucine at 1010 (T1010I), methionine-to-threonine at 1268 (M1268T), tyrosine-to-histidine at 1248 (Y1248H), and deletion of Sema domain ((ΔSema); as well as on pCMV6-ERBB2-DDK for deletion of L domain 1 (ΔD1), furin-like domain (ΔD2), L domain 3 (ΔD3), growth factor receptor domain IV (ΔD4) and tyrosine kinase domain (ΔTK) using QuikChange Lightning Kit (Agilent, #210519) according to the manufacturer's protocol. For transfection, cells were incubated with plasmids and FuGENE HD (Promega, #E2311) in Opti-MEM medium according to the manufacturer's recommendation. For transient RNA interference, *MET* siRNAs, *ERBB2* siRNAs, and AllStar-negative control siRNA were obtained from Qiagen (Supplementary Table 3). siRNA transfection was conducted with JetPRIME reagent (Polyplus Transfection) according to the manufacturer's recommendation.

**Transwell invasion and migration assays.** Matrigel invasion assay was carried out with Corning® BioCoat™ Matrigel Invasion Chambers (#354480). In brief, 25,000–50,000 cells were plated in the upper chamber of the transwell plate with serum-free medium, with the lower chambers containing complete medium. For gene-silencing experiments, cells were incubated with siRNA overnight prior to plating. Tumor cells were allowed to invade for 24–36 h before the transwell inserts were fixed with 4% paraformaldehyde, stained with gentian violet, and imaged for analysis. Scratch (migration) assay was performed by creating a gap on a cell monolayer, and images were captured (4× for invasion; 10× for migration) using a brightfield microscope at the indicated time intervals. ImageJ Software (NIH) was used to quantify the count of invaded cells and percentage of wound closure, with the data expressed as ± SD.

**Anchorage-independent colony-forming assay.** The colony-forming assay required three layers of soft agar/medium matrix in 24-well plate: bottom layer (0.6 ml) containing complete medium and 0.6% agar; a middle layer (0.5 ml) containing complete medium, 0.36% agar and cells (5000–10,000 cells/well); and a top layer (0.5 ml) comprising complete medium with either vehicle or the indicated inhibitors of different concentrations. Culture medium was changed weekly. For gene-silencing experiments, cells were incubated with siRNA overnight prior to culturing in soft agar. After 4 weeks of culture, colonies were stained with MTT stain (Promega, #G4000) for 2 h, and imaged. Colonies were counted with ImageJ Software, and expressed as mean ± SD relative to respective controls.

**Animal models.** All mouse models were established in 8- to 10-week-old female SCID mice by adhering to the Institutional Animal Care and Use Committee (IACUC) guidelines on animal use and handling. The experiments were approved by the Singhealth IACUC. The PDX is approved by SingHealth Centralised Institutional Review Board (CIRB). For in vivo metastases model, $1 \times 10^6$ of cancer cells (empty vector control, MET$^{wt-tGFP}$ and MET$^{N375S-tGFP}$) were injected intravenously into tail vein of each mice. For tumor burden analysis, mice were sacrificed at 10 weeks to harvest the lungs. The lungs were imaged after staining with Bouin's solution (Sigma, #HT10132), fixed in 10% (vol/vol) neutral buffered formalin (NBF) overnight, and embedded in paraffin. To evaluate the metastatic burden, FFPE lung tissues were sectioned and stained with hematoxylin and eosin (H&E), and analyzed using ImageJ Software as previously described[42]. Metastatic burden is estimated by the coverage of metastatic tumor volume in total lung volume using stereological quantification, expressed as mean percentage ± SD. A total of three lung lobes per mice were used for quantification. For survival analyses, mice were orally administered with vehicle [seven parts 30% (w/v) Captisol (Cydex) and three parts 30% (w/v) PEG300 (Sigma)] or 15 mg/kg afatinib daily (QD). Mice were randomized to treatment or control strata according to their body weight. Kaplan–Meier survival was measured using GraphPad Prism with statistical significance calculated based on log-rank test.

For xenograft study, $5 \times 10^6$ isogenic H2170 cells (empty vector control, MET$^{wt-tGFP}$ and MET$^{N375S-tGFP}$) in 100 μL of PBS or 1–2 mm$^3$ dices of PDX tumors (HCC26-0808B and HCC13-0109) were injected subcutaneously into both flanks of each mice. Tumor size was monitored and measured three times weekly with Vernier calipers (tumor volume = length × width$^2$ × 3.14159/6). For inhibitor studies, mice were randomly assigned into stratified groups ($n = 10$ tumors/5 mice per group), and treatment was initiated when the tumors reached 150 mm$^3$. Crizotinib, INC280, ASLAN001, lapatinib, and afatinib were formulated with seven parts 30% (w/v) Captisol (Cydex) and three parts 30 % (w/v) PEG300 (Sigma), and given orally (p.o.). Pertuzumab and trastuzumab were formulated in 0.9% NaCl and given intraperitoneally (i.p.). Crizotinib (50 mg/kg), INC280 (25 mg/kg), lapatinib (75 mg/kg), and afatinib (15 mg/kg) were given daily (QD); ASLAN001 (50 mg/kg)

was given twice daily (BID); trastuzumab and pertuzumab were given at 10 mg/kg twice weekly (Q3.5D).

**Protein array, immunoblotting, and cellular fractionation.** The Human Proteome Profiler Phospho-kinase antibody array (R&D Systems) and sodium dodecyl sulfate–PAGE (SDS-PAGE) was performed. Whole-cell lysates were extracted with cell lysis buffer (Sigma, #2978) supplemented with protease and phosphatase inhibitors (Thermo Fisher, #78440). Cell lysates (250 μg) were mixed with biotinylated detection antibodies for phospho-kinase antibody array; 30–40 μg of proteins were used for immunoblotting. Signal detection was performed with Chemiluminescence detection system (Avansta, #K-12045-D50). Densitometric data from The Phospho-kinase antibody array was calculated with Quick Spots image analysis software (Western Vision Software) as mean pixel density. Fold change for each target spots was analyzed in relative to MET$^{wt-tGFP}$ cells, and presented as fold regulation (positive fold regulation indicates relative fold increase in MET$^{N375S-tGFP}$ cells; negative fold regulation indicates relative fold decrease in MET$^{wt-tGFP}$ cells). Cellular fractionations were performed according to the manufacturer's protocol (Active Motif, #40010 for nuclear/cytosolic fractionation; Thermo Fisher, #89842 for membrane/cytosolic fractionation). Protein concentration was determined by BCA protein assay (Life Technologies, #23227) to normalize protein quantity for immunoblotting.

**Immunoprecipitation.** Whole-cell lysate (1 mg) from the indicated samples were prepared and normalized as per described in immunoblotting. For inhibitor and rSema studies, indicated compounds or rSema were added 24 h before cells were harvested. Normalized lysates were incubated with 5 μg of primary or anti-IgG antibodies overnight at 4 °C with gentle agitation. Sample–antibody mixture was incubated with magnetic protein G Dynabeads™ (Thermo Fisher Scientific, #10003D) for co-immunoprecipitation, and washed three times with protein binding buffer (150 mM NaCl, 20 mM Tris pH 8.0, 1% NP-40, and supplemented with protease and phosphatase inhibitors). The immunoprecipitant was eluted in sample buffer containing 1% β-mercaptoethanol and subjected to SDS-PAGE immunoblotting. For HGF and serum-depletion study, cells were starved overnight with 0.1% FBS, and induced by 10 ng/ml HGF for 30 min.

**Duolink proximity ligation assay (PLA).** Cultured cells on coverslips were fixed with 4% paraformaldehyde at room temperature for 30 min. Duolink PLA staining was conducted according to the manufacturer's protocol (Duolink, #DUO92101). Fluorescence imaging of PLA (Texas Red) and DAPI signal was performed under ×40 (cultured cells). Image deconvolution and analysis were performed using ImageJ software (NIH). The mean PLA/cell intensity ± SD from at least 50 cells in each group was tabulated.

**Immunofluorescence (IF) staining.** Immunofluorescence staining was performed on cultured cells fixed with 4% paraformaldehyde. The fixed cells were blocked and permeabilized with 2% BSA, 0.1% Triton X-100 in TBS. Immunostaining was performed with anti-MET (mouse) and anti-HER2 (rabbit) overnight at 4 °C, followed by incubation with Alexa Fluor-488 or -594 secondary antibodies (1:1000, Molecular Probes, Life Technologies, A28175 and A-11012) for 2 h at 25 °C, and by DAPI nuclear staining for 10 min. Confocal images (five fields) were acquired with LSM880 (Zeiss Germany) with ×40 objective. Image analysis was conducted with ImageJ Software, where the HER2 fluorescence signal was overlaid onto the MET channel, and the percentage of MET that co-localized with HER2 was tabulated ± SD.

**Immunohistochemistry (IHC) staining and analyses.** Tumors extracted from xenograft studies and biopsies were fixed overnight in formalin after surgical excision, and embedded in paraffin and sectioned (4 μm). Tumor sections were deparaffinized using standard histologic procedures prior to antigen retrieval. IHC was performed using standard staining protocol, and color development was performed using EnVision+ System-HRP (DAB) kit (Dako, Agilent Technologies) according to the manufacturer's recommendation. Cell nuclei were counterstained with Gill 3 Hematoxylin (Thermo Scientific, #72604). Slide images were captured under optical imaging microscope (×20 objective for xenograft tumor, ×10 for biopsy tumor). Images from five xenograft tumor samples for every treatment group were analyzed with ImageJ software with the Color Deconvolution Plugin. IHC staining was analyzed as the percentage of DAB-positive cells/100 nuclei ± SD for at least 20 fields.

**Statistic and data reporting.** All experiments were independently conducted three times unless stated otherwise. The results are presented as mean ± SD. Statistical analysis for the comparison between two groups was conducted using Student's unpaired *t* test, while comparisons between multiple groups was conducted using two-way ANOVA with Bonferroni Post hoc tests.

**Reporting summary.** Further information on research design is available in the Nature Research Reporting Summary linked to this article.

## Data availability

The RNAseq data sets generated during this study have been deposited in GEO with the accession number GSE128956. The targeted exome sequencing datasets analyzed during the current study have been deposited in BioProject with the accession number PRJNA529714. All other data are available in the Article file, Supplementary Information, or available from the authors upon reasonable request. The source data underlying Figs. 3C, D, 4C, D, G, J–M, 5A–D, and 6M are provided as a Source Data file.

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

## Acknowledgements

This work was funded by the Singapore Ministry of Health's National Medical Research Council (NMRC/CSA-SI/0006/2016), the National Research Foundation (NRF) Singapore and the Singapore Ministry of Education under its Research Centres of Excellence initiatives (Boon-Cher Goh); as well as to Boon-Cher Goh and Li-Ren Kong by Singapore Ministry of Health's National Medical Research Council under its NCIS Centre Grant (NMRC/CG/012/2013). Chandra S Verma and Srinivasaraghavan Kannan thank A*STAR, NRF and Singapore Economic Development Board for grant support.

## Author contributions

Conceptualization, L.R.K. and B.C.G.; methodology, L.R.K., N.A.B.M.S., R.W.O., C.W.F., J.S.J.L., B.J.Y., M.E.N., D.K., and H.T.H.; software, T.Z.T., S.K., and C.S.V.; validation, L.R.K. and N.A.B.M.S.; formal analysis, L.R.K., N.A.B.M.S., T.Z.T., C.W.F., S.K., C.S.V., R.M.G., Y.C.L., and D.K.; investigation, L.R.K., N.A.B.M.S., R.W.O., J.H., Y.H., S.C.L., D.K., and B.C.G.; resources, D.S.W.T., N.G.I., R.S., J.H., Y.H., S.G.L., S.C.L. H.T.H. and B.C.G.; data curation, L.R.K., T.Z.T., and S.C.L.; writing—original draft, L.R.K. and B.C.G.; writing—review and editing, L.R.K., N.A.B.M.S., N.L.S., R.M.G., H.P.K., S.C.L., D.K., and B.C.G.; visualization, L.R.K., N.L.S., and R.M.G.; supervision, B.C.G.; project administration, L.R.K. and B.C.G.; funding acquisition, L.R.K. and B.C.G.

## Competing interests

S.K. and C.S.V. are the founder directors of Sinopsee therapeutics, a biotech company developing molecules for therapeutic purposes; this work has no conflict with the company. The remaining authors declare no competing interests.
