## [Peer Review File · Nature Communications]

Reviewers' comments:

Reviewer #1 (Remarks to the Author): Expert in MET signalling

Kong et al report a mechanism of action for the relation between a germinal MET polymorphism and the aggressiveness of SCC tumors. Ten patient cohorts were analyzed for the presence of the METN375S which was subsequently linked to the survival of the patients. The SCC cohorts and a gastric cohort showed a significantly shorter RFS in the presence of the mutation. The functionality of the METN375S on migration, invasion and proliferation is tested in isogenic cell lines as well as CRIPR/CAS knock-ins. A variety of methods (comparative gene expression, phosphoprotein array, SILAC labeling, PLA, immunoprecipitation) is subsequently used to define the interaction protein for the METN375S as well as the interaction domain. The mutation in the METN375S semaphorin domain provides an interaction site for HER2 resulting in enhanced phosphorylation and signaling of HER2. Inhibition of HER2 predominantly through the use of small molecules results in inhibition of tumor cell growth in vitro and in vivo. The METN375S interaction seems to be specific for SCC since HER2 targeted agents are ineffective in two PDX models of HCC METN375S. Finally, two clinical cases are described of two refractory HNSCC patients that benefit of lapatinib treatment by PR or tumor reduction.

I find the results interesting since they provide insight in a novel oncogenic MET signaling pathway and warrant the exploration between METN375S and HER2 targeting in SCC patients that become refractory.

The abstract, introduction and discussion are very well described. Basically, the results section of the paper can be divided into three parts:

1. METN375S expression and RFS and functionality in terms of tumor burden
2. Determination of the binding partner for METN375S including domain analysis
3. Effectivity of HER2 inhibition in the (pre)-clinical setting

In the results section the first and third sections of the work are very solid and well described. The second section misses a logical flow, bias to H2170 cells and some confusing experimental details. The logical flow to the discovery of HER2 as the binding partner for METN375S is lacking. The comparative gene analysis points into the direction of cell/cell interaction, the phospho protein array lacks HER2 phosphorylation and HER2 was among the lower top membrane candidates in the SILAC labeling. The description of the immunoprecipitations with GFP is confusing since tGFP is used whereas the isogenic cell lines have not been described as a GFP fusion. Due to the extensive array of methods used, the interactions studied and the missing link from the analysis to HER2 as a binding partner and the fact that the majority of the work has been performed with the H2170 cell line paper reads unbalanced to the reviewer.

Thus I recommend revision of the article by limiting the inclusion of data that do not support the finding. To strengthen the observations with less bias, two PDX models like used for S9 (D-E) are recommended.

General comments:

1. Although cMET is involved in migration and metastasis, the experiments performed in this manuscript do not support that the METN375S/HER2 interaction drives metastasis. In the in vitro experiments, no proliferation inhibitors like mitomycin C are added during the migration or invasion assay. None of the in vivo studies shows enhanced metastasis. As such suggested metastasis part in S13 is incorrect.
2. Incorrect statement in the discussion. The HER2/HER3 dimer mediates the strongest signaling function and is far more active in comparison to HER2/HER2 or HER2/EGFR. Please include those references.

Comments on text/figures

- S2 to demonstrate expression levels of MET a FACS analysis is preferred over western blots
- Figure 3A, overexpression long exposure unconvincing
- S6DE Interaction with cMETN375S not convincing since interaction with wild type visible in both cancer cell types
- 4HI not convincing, that rSemaN375S domain exhibits stronger growth inhibitory effects compared in cMETN375S compared to WT.
- S7 misses pEGFR (lapatinib equally effective to HER2 and EGFR) inconsistent with S9 does not consistent
- Supplemental figure 2D relates to wound closure according to the graph, whereas the legend addresses colony formation
- throughout the document spelling mistakes were observed, to mention a few "tyrosine kinas domain" (line 22 page 14) "herrceptin"(line 4 page 29)
- Fig. 6N should be Fig. 6O (line 16 and 17, page 11) and vice versa
- Line 3 page 15, a word is missing after HER2 (signaling, phosphorylation)
- In all figures the u (micro) symbol is difficult to read

Reviewer #2 (Remarks to the Author): Expert in HNSCC

For: Nature Communications

Kong et al. "A common MET polymorphism harnesses HER2 signaling to drive aggressive squamous cell carcinoma"

Ref: NCOMMS-19-15859-T

Corresponding Author: Boon Cher Goh

The manuscript "A common MET polymorphism harnesses HER2 signaling to drive aggressive squamous cell carcinoma" by Kong et al. describes the N375S polymorphism of the MET gene to heterodimerize with HER2 and with that to drive an aggressive phenotype of squamous cell carcinoma.

The authors describe METN375S to be a prognostic marker for HNSCC and LUSC, although polymorphism frequency is not altered in cancer patients. Cell migration, invasion, colony formation and metastases formation was increased while elevated levels of Src-phosphorylation was detected in METN375S cells. Additionally METN375S was found to interact with HER2 which is described to drive the aggressive phenotype. In this context HER2 inhibition but not MET inhibition was efficient, which was utilized when treating HNSCC patients with afatinib.

The manuscript describes a very interesting and potentially clinically relevant interaction between MET and HER, identifying METN375S as a potential predictive biomarker for the use of HER2 targeting for SCC. Since biomarker identification is vital for the further progress in individualized SCC therapy - which is still urgently needed - the article hits a crucial point.

Nevertheless, there are still some open questions and some concerns which have to be addressed to justify publication in Nature Communications.

Major Concerns

1. The authors state, that METN375S expression is relevant for SCC of the lung and the head and neck. For isogenic experiments only lung SCC cell lines were used but finally HNSCC patients were treated. Please provide some key experiments also for isogenic HNSCC cell lines.
2. Please provide protein standards for all Western blots (WB). It is not possible to check the MW of the indicated proteins.

3. Fig. 2 and S2: Please provide WB analyses controlling MET and pMET all the different cells used (EV, METwt and METN375S) including protein standards.
4. Fig. S3I & S7C indicates no dramatic difference in MET expression in H2170 EV cells compared to METwt. Only METN375S cells display clearly elevated levels of MET. Furthermore, there seems to be a difference in the presence of pro-MET (upper lane, or is this MET-GFP?). While no pro-MET is detected in the EV control, relative stronger signals are detectable in the METwt compared to the METN375S. Such conclusion can also be drawn from Fig. 3D. Please comment on this and discuss, if these observations might influence the results.
In this context, it is not clear, if all MET constructs contain GFP (compare page 19, line 10 and line 17). Please clarify and indicate precisely in the WB.
5. The authors showed increased cell migration, invasion, colony formation and metastases formation upon METN375S expression. None of these parameters have been addressed in Fig. S3. Therefore, the authors have not shown, that MET inhibition is not affective. Please provide data on cell migration, invasion, colony formation or metastases formation after MET inhibition.
6. PLA data are not convincing: PLA signals seem to be localized in the perinuclear region, indicating ER localization. MET and HER2 should be preferentially located at the plasma membrane. Therefore please provide additional confocal analysis of METN375S and HER2 (co-)localization. So far the data presented only show interaction of HER2 and METN375S, however it remains unclear, if this is an indirect or direct interaction. Therefore, statements such as "METN375S to heterodimerize with HER2" (Abstract) should be toned down or additional data should be provided, validating a direct binding .

Minor concerns:

1. Page 4, lane 26: How was amplicon-enriched NGS performed? There is no protocol in the manuscript and not in the given reference 19.
2. Fig. 3E: pMET and pHER2 IHC are depicted, but the text indicates pSrc instead of pHER2 (page 6, lane 5). Please clarify.
3. Page 7, lane 2: The authors claim a hyperactive signaling of METN375S. Since only Src phosphorylation seems to be increased and MET inhibition has no significant influence, there is no obvious hyperactivity. Please use a moderate wording.
4. There are no WB to control the knockdown shown in Fig. 2B, F, D. Please provide.
5. Since there is no detailed analysis of nuclear MET, there is no real benefit from Fig. 4A, please shift it to supplementary information. Furthermore, the authors have shown, that crizotinib has no effect in terms of proliferation and tumor growth. Therefore I cannot follow the conclusions drawn from this experiments (page 7, lane 8); crizotinib likely does not induce stress to the cells.
7. The observation, that crizotinib interrupts interaction with HER but does not influence the tumor growth or cell invasion might argue for additional players or interaction partners, who might be responsible for ongoing HER2 activity. Please discuss.

Reviewer #3 (Remarks to the Author): Expert in kinase biochemistry and structure

This manuscript by Goh and colleagues describes a common germline polymorphism of Met, N375S, that confers a more aggressive phenotype and poorer prognosis to squamous cell carcinomas, apparently through heterodimerization of the variant receptor with the Her2/Neu kinase. In cell lines and in xenografts, Her2 inhibitors (both antibodies and small molecules) are effective in inhibiting growth and downstream signaling of cells expressing MetN375S, whereas Met inhibitors have little if any effect in these cells. This effect can only be demonstrated for SCC (head and neck, lung), and not other tumors expressing MetN375S. Pilot clinical data for two patients with refractory MetN375S tumors treated with Her2 inhibitors are encouraging.

This is interesting a potentially quite important work, in identifying a novel therapeutic approach to treat tumors with a relatively common Met polymorphism. The mechanistic implications are also quite interesting, in that activating heterodimerization of receptor tyrosine kinases from different families is not a well-established activation mechanism. Unfortunately the precise mechanism of activation remains a bit mysterious, as Her2 activation seems to be maintained and even enhanced by Met inhibitors in MetN375S cells; whether a kinase-inactive N375S mutant (as opposed to a kinase domain deletion mutant) could still promote Her2 activation would be informative in this regard. However despite a number of remaining questions regarding mechanism, this work provides important new insights and sets the stage for future studies to explore clinical efficacy and molecular mechanism.

Specific points:

1. I found myself confused regarding the various cell models used for particular experiments. Some use ectopic overexpression, and some specific knock-in of the N375S variant into the endogenous locus, and in some cases apparently a GFP fusion is used, but it is not always clear what cells were being used for particular experiments, and also whether the knock-in was homozygous or heterozygous for the variant allele.
2. For Fig. 3A and B, some more explanation is needed (e.g. how is "fold regulation" calculated; is this in fact a receptor tyrosine kinase antibody array, or more likely a phosphoprotein antibody array, etc.)
3. On p. 12 and p. 14, the authors state that MetN375S phosphorylation is necessary for Her2 activation, but this seems in direct contradiction to their data with Met inhibitors, e.g. Fig. 5D, where Met phosphorylation is low yet Her2 phosphorylation is high. Further, there appears to be no heterodimerization of Met and Her2 in the presence of Met inhibitors. While I realize all the mechanistic details may not be worked out, some plausible mechanism(s) should be briefly discussed. Much is known from structural studies about the mechanisms of EGFR family activation via dimerization, which may be relevant here.

Reviewer #4 (Remarks to the Author): Expert in kinase biochemistry and structure

The paper from Kong and colleagues is an impressive piece of work that describes a novel and unexpected alternative signaling mechanisms of a polymorphism (N275S) in the MET receptor tyrosine kinase in patients with certain head/neck and lung carcinomas. Based on solid epidemiological and clinical evidence on inferior outcome of patients with MET N275S, the authors use cellular models and mouse xenografts to show higher migration, transformation and metastatic potential of MET N275S. Biochemical analysis shows plausible downstream signaling differences when compared to MET wt. Careful pharmacological perturbation studies show low sensitivity for several MET TKIs in MET N275S cells and lead to the hypothesis that heterodimerization with HER2 may happen. Several well-controlled and state-of-the-art methods, including differential quantitative proteomics, PLA assays and amply controlled co-IPs in different cell lines unequivocally show HER2-MET N275S heterodimerization. Based on these findings, perturbation of MET N275S signaling with HER2 TKIs and antibodies results in inhibition of cell growth and signaling in cellular models in vitro, as well as in inhibition of tumor growth and prolongation of survival in mouse xenograft models. Finally, clinical efficacy of the proposed novel treatment approach is demonstrated in two patients.

Overall, this is an exceptional paper in terms of amount and quality of data, clarity of presentation and how rigorous, systematic and well controlled the experiments were done.

(Very) minor points:

1. I was wondering about the relatively high (from 1-20 microM) GC50 values of the MET inhibitors in Figure S3. Do the authors think that the weak inhibitory activity is on-target? Given that authors describe CRISPR-Cas9 editing of these cell lines to make knock-ins (Fig. S2, page 5), have they

also done CRISPR knock-outs for MET? If such MET knock-out cell lines would still show low sensitivity to MET TKIs, this would indicate off-target activity of the drugs (and further strengthen the data). Please comment.

2. Some typos to correct:

page 14, line 22: "kinase", not "kinas"

page 15, line 3: I think one word is missing. Either "leads to constitutively active HER2" or "leads to constitutive HER2 signaling"

"leads to constitutively active HER2" or

page 15, line 30: "It is of note" not "In is of note"

Reviewers' comments:

Reviewer #1 (Remarks to the Author): Expert in MET signalling

Kong et al report a mechanism of action for the relation between a germinal MET polymorphism and the aggressiveness of SCC tumors. Ten patient cohorts were analyzed for the presence of the METN375S which was subsequently linked to the survival of the patients. The SCC cohorts and a gastric cohort showed a significantly shorter RFS in the presence of the mutation. The functionality of the METN375S on migration, invasion and proliferation is tested in isogenic cell lines as well as CRIPR/CAS knock-ins. A variety of methods (comparative gene expression, phosphoprotein array, SILAC labeling, PLA, immunoprecipitation) is subsequently used to define the interaction protein for the METN375S as well as the interaction domain. The mutation in the METN375S semaphorin domain provides an interaction site for HER2 resulting in enhanced phosphorylation and signaling of HER2. Inhibition of HER2 predominantly through the use of small molecules results in inhibition of tumor cell growth in vitro and in vivo. The METN375S interaction seems to be specific for SCC since HER2 targeted agents are ineffective in two PDX models of HCC METN375S. Finally, two clinical cases are described of two refractory HNSCC patients that benefit of lapatinib treatment by PR or tumor reduction.

I find the results interesting since they provide insight in a novel oncogenic MET signaling pathway and warrant the exploration between METN375S and HER2 targeting in SCC patients that become refractory.

The abstract, introduction and discussion are very well described. Basically, the results section of the paper can be divided into three parts:

1. METN375S expression and RFS and functionality in terms of tumor burden
2. Determination of the binding partner for METN375S including domain analysis
3. Effectivity of HER2 inhibition in the (pre)-clinical setting

In the results section the first and third sections of the work are very solid and well described. The second section misses a logical flow, bias to H2170 cells and some confusing experimental details. The logical flow to the discovery of HER2 as the binding partner for METN375S is lacking.

We thank the reviewer for the encouraging comments. In order to improve on the logical flow to the discovery of HER2 as the interacting partner for MET-N375S, we have re-organized the manuscript by including/excluding several pieces of data. In summary, we first characterized the oncogenic properties of MET-N375S, and showed that the increase in cellular motility is not regulated at the transcriptional level, thus leading to the discovery of MET-HER2 dimerization as the key mechanism of the observed phenotype.

The comparative gene analysis points into the direction of cell/cell interaction,

We agree with the reviewer that the gene expression analyses are not beneficial to the discovery of HER2 as a binding partner of MET^{N375S}. We have revamped Supplementary Figure 4, removed the pathway enrichment analyses and retained only the EMT score (Pg 7, line 27-30), which emphasizes on a transcriptional-independent mechanism leading to enhanced cell motility in N375S cells.

the phospho protein array lacks HER2 phosphorylation

We agree with the reviewer on this, but sadly the kinase array used in our study does not include p-HER2, which was not taken into consideration at the initial discovery stage of the study.

and HER2 was among the lower top membrane candidates in the SILAC labeling.

We are aware that HER2 seems to be among the SILAC hits with low confidence from the tabulation in Figure 4B. In order to validate HER2 as a true positive hit, we had performed coIP on all the potential binding partners identified from SILAC (Supplementary Figure 5A) and indeed none of the tested targets (RAPH1, BAG3, ENO1, RIT1) could be co-immunoprecipitated with both MET variants. We reasoned that the other SILAC targets were likely false positive signals.

The description of the immunoprecipitations with GFP is confusing since tGFP is used whereas the isogenic cell lines have not been described as a GFP fusion.

We apologize for the confusion caused by our labeling of the various cell lines. MET plasmid used to generate the isogenic clones contain a turbo-GFP tag which was described in the Methods (Pg 20, line 18). However, these details were not emphasized in the initial submission, which we have rectified throughout the manuscript. For isogenic cell lines overexpressing MET, we have named them (MET^{wt-tGFP} and MET^{N375S-tGFP}), whereas nomenclature of MET^{wt} and MET^{N375S} are now referring to the respective c-MET isoforms. We hope that this will increase the clarity and improve the description of the manuscript.

Due to the extensive array of methods used, the interactions studied and the missing link from the analysis to HER2 as a binding partner and the fact that the majority of the work has been performed with the H2170 cell line paper reads unbalanced to the reviewer.

We will like to thank the reviewer for this comment, and we agree with the reviewer on the “unbalanced” focus on H2170 cells in the main text. However, we will like to emphasize that several cell lines have been used in the study to compliment H2170 cells and to avoid biasness, such as the mesenchymal Calu-1, the patient derived NPC7, as well as the CRISPR Knock-in cells (H2170 and Calu-1). The manuscript focused on the discussing H2170 intentionally to avoid confusing the readers (as pointed by Reviewer 3).

Thus I recommend revision of the article by limiting the inclusion of data that do not support the finding. To strengthen the observations with less bias, two PDX models like used for S9 (D-E) are recommended.

We will like to thank the reviewer for the suggestions. We have made multiple changes to the figure and provided clearer rationale on the experimentations. Also, we have screened our collection of patient-derived cell lines and xenografts to select the suitable models for the study (NPC7 cell line, and the two PDXs). In general, we feel that the positive responses in the two clinical cases that supported our findings would be convincing validation.

In summary,

- 1) We have removed the less informative gene expression analyses (Supplementary Figure 4B and 4C)
- 2) We have removed the nuclear-cytosolic fractionation of MET protein in MET^{wt-tGFP} and MET^{N375S-tGFP} cells (Figure 4A).

General comments:

1. Although cMET is involved in migration and metastasis, the experiments performed in this manuscript do not support that the METN375S/HER2 interaction drives metastasis. In the in vitro experiments, no proliferation inhibitors like mitomycin C are added during the

migration or invasion assay. None of the *in vivo* studies shows enhanced metastasis. As such suggested metastasis part in S13 is incorrect.

We thank the reviewer for this suggestion. In our experimental setting, we did not include an inhibitor of cell proliferation as we have concern over its synergistic/additive effect with the various compounds used in the study (crizotinib, afatinib etc). Nonetheless, we agree that it is important to demonstrate that enhanced cellular motility observed in Figure 2 is independent of growth rate in various cell type. We have conducted growth curve to both H2170 and Calu-1 cells harboring WT and N375S MET as shown below (Reviewer Figure 1). The data suggest that the 2D growth of MET^{N375S-tGFP} is instead slower than that of MET^{wt-tGFP}, therefore the enhanced cell migration of MET^{N375S-tGFP} in wound healing assay is unlikely due to increase in cell proliferation. Instead, expression of MET^{N375S} significantly increased growth of 3D colony and *in vivo* tumors, again emphasizing the malignant transformation associated with the MET variant.

We will also like to draw attention to the observation that *in vivo* tail vein injection of MET^{wt} and MET^{N375S} cells exhibited differential lung metastases as described in Pg 6, line 15-19 “In addition, while tail vein engraftment of both MET^{wt-tGFP} and MET^{N375S-tGFP} Calu-1 clones developed significant lung metastases compared to EV control (Fig. 2H), MET^{N375S-tGFP} clones demonstrated enhanced metastatic potential by forming large ‘cannonball’ metastatic nodules compared to MET^{wt-tGFP} (Fig. 2H), with a greater tumor burden (Fig. 2I).” We believe these observations are supportive of the role of MET^{N375S} in driving metastasis in SCC cells.

Reviewer Figure 1: Growth rate of H2170 clones expressing WT and N375S-MET. 5,000 cells were seeded and viable cell count was measured with Cell Titre Glo assay (at 0, 6, 24, 72 and 96 hr). Data is presented as mean \pm SD (n = 2).

2. Incorrect statement in the discussion. The HER2/HER3 dimer mediates the strongest signaling function and is far more active in comparison to HER2/HER2 or HER2/EGFR. Please include those references.

We thank the reviewer for pointing this out. We have amended the statement to “HER2 mediates a potent downstream signaling when activated through its highly catalytic TK domain” (Pg 14, line 32).

Comments on text/figures

- S2 to demonstrate expression levels of MET a FACS analysis is preferred over western blots

In this study, we have utilized Western blotting as a universal technique to check for MET-tGFP expression (for isogenic and CRISPR cells). As each MET clones is established through single cell clonal selection, we think that FACS analysis to determine MET expression will not provide additional value to the analysis. Moreover, the subsequent comparisons of phosphoproteins in MET^{wt-tGFP} and MET^{N375S-tGFP} cells were conducted with Western blotting. Nonetheless we have conducted the experiment as suggested by the reviewer (Reviewer Figure 2). As shown below, both MET^{wt-tGFP} and MET^{N375S-tGFP} are sorted as a single peak (indicative of single cell clone), while MET^{wt-tGFP} have slightly higher tGFP expression as compared to MET^{N375S-tGFP}, which is concordant with our WB analyses (Figure 2).

Reviewer Figure 2: FACS analyses of tGFP expression in H2170 MET^{wt} in MET^{N375S-tGFP} cells.

- Figure 3A, overexpression long exposure unconvincing
We have replaced the long exposure blot (Figure 3A right) with one that is more representative of the densitometric quantitation (Figure 3B).
- S6DE Interaction with cMETN375S not convincing since interaction with wild type visible in both cancer cell types
We have the same observations as the reviewer, and will like to discuss this further. We agree that forced expression of both exogenous MET variants showed co-immunoprecipitation with HER2 in both cell lines tested. This is likely due to the inherent binding affinity of MET to HER2, as reported by Yang L, et al. (Sci. Transl. Med. 11, eaav1620 (2019)), which could be reinforced under transient overexpression of MET. However, N375S exhibits way stronger

interaction to HER2 in our stable MET-tGFP expressing cells, suggesting that N375S is a stronger binding partner of HER2 with higher affinity compared to its WT counterpart. This is again reiterated by our structural simulation (Figure S14) that N375S mutation modified the Sema domain of MET to expose more interacting surface with HER2. This has been discussed in Pg 15, line 15-19.

- 4HI not convincing, that rSemaN375S domain exhibits stronger growth inhibitory effects compared in cMETN375S compared to WT.

To increase clarity of the data presentation, we have supplemented Figure 4H and 4I with the mean values of each bar. In addition, we will like to clarify that the claim made “While both rSema proteins dose-dependently reduced the cell viability of MET^{wt-tGFP} cells (Fig. 4H), rSema^{N375S} exhibited stronger growth inhibitory effects in MET^{N375S-tGFP} cells compared to rSema^{wt} (Fig. 4I).” (Pg 9, line 14-16) was referring to the strong inhibitory effect of rSEMA^{N375S} compared to rSEMA^{wt} in MET^{N375S-tGFP} cells, and not comparing the effects of rSEMA^{N375S} on MET^{wt-tGFP} and MET^{N375S-tGFP} cells.

- S7 misses pEGFR (lapatinib equally effective to HER2 and EGFR) inconsistent with S9 does not consistent

The blots for p-EGFR have been supplemented in Figure S7 in the revised manuscript.

- Supplemental figure 2D relates to wound closure according to the graph, whereas the legend addresses colony formation

We apologize for the mistake. The figure legend is correctly labelled, and we have amended the y-axis of the graph (Figure 2D).

- throughout the document spelling mistakes were observed, to mention a few “tyrosine kinas domain” (line 22 page 14) “herrceptin”(line 4 page 29).

These spelling mistakes have been amended. We have also proof-read the other part of the revised manuscript to prevent for additional spelling errors.

- Fig. 6N should be Fig. 6O (line 16 and 17, page 11) and vice versa

These mistakes have been amended.

- Line 3 page 15, a word is missing after HER2 (signaling, phosphorylation)

We have amended the statement to “constitutively active HER2” Pg 16, line 2.

- In all figures the u (micro) symbol is difficult to read

We apologize for this oversight and lack of consideration. We have edited the figures to improve on the data presentation.

Reviewer #2 (Remarks to the Author): Expert in HNSCC

For: Nature Communications

Kong et al. “A common MET polymorphism harnesses HER2 signaling to drive aggressive squamous cell carcinoma”

Ref: NCOMMS-19-15859-T

Corresponding Author: Boon Cher Goh

The manuscript “A common MET polymorphism harnesses HER2 signaling to drive

aggressive squamous cell carcinoma” by Kong et al. describes the N375S polymorphism of the MET gene to heterodimerize with HER2 and with that to drive an aggressive phenotype of squamous cell carcinoma.

The authors describe METN375S to be a prognostic marker for HNSCC and LUSC, although polymorphism frequency is not altered in cancer patients. Cell migration, invasion, colony formation and metastases formation was increased while elevated levels of Src-phosphorylation was detected in METN375S cells. Additionally METN375S was found to interact with HER2 which is described to drive the aggressive phenotype. In this context HER2 inhibition but not MET inhibition was efficient, which was utilized when treating HNSCC patients with afatinib.

The manuscript describes a very interesting and potentially clinically relevant interaction between MET and HER, identifying METN375S as a potential predictive biomarker for the use of HER2 targeting for SCC. Since biomarker identification is vital for the further progress in individualized SCC therapy - which is still urgently needed - the article hits a crucial point.

Nevertheless, there are still some open questions and some concerns which have to be addressed to justify publication in Nature Communications.

Major Concerns

1. The authors state, that METN375S expression is relevant for SCC of the lung and the head and neck. For isogenic experiments only lung SCC cell lines were used but finally HNSCC patients were treated. Please provide some key experiments also for isogenic HNSCC cell lines.

Regarding the selection of cell lines, we have performed key experiments on a patient-derived heterozygous MET^{N375S} cell line (NCC-NPC7) (Supplementary Figure S9), which support our findings. Some of the key experiments included sensitivity assays to crizotinib/afatinib treatment, responses to MET/HER2 inhibition by immunoblotting and MET/HER2 co-immunoprecipitation assays. Currently, our clinical trial studying HER2 inhibition in MET^{N375S} positive patients is open for recruitment of both lung and head and neck SCC patients.

2. Please provide protein standards for all Western blots (WB). It is not possible to check the MW of the indicated proteins.

The MW of the indicated proteins have been included in all Western blot data.

3. Fig. 2 and S2: Please provide WB analyses controlling MET and pMET all the different cells used (EV, METwt and METN375S) including protein standards.

The expressions of MET and/or p-MET have been supplemented in all the relevant WB in Figure 2 and Supplementary Figure 2.

4. Fig. S3I & S7C indicates no dramatic difference in MET expression in H2170 EV cells compared to METwt. Only METN375S cells display clearly elevated levels of MET. Furthermore, there seems to be a difference in the presence of pro-MET (upper lane, or is this MET-GFP?). While no pro-MET is detected in the EV control, relative stronger signals are detectable in the METwt compared to the METN375S. Such conclusion can also be drawn from Fig. 3D. Please comment on this and discuss, if these observations might influence the results.

In this context, it is not clear, if all MET constructs contain GFP (compare page 19, line 10 and line 17). Please clarify and indicate precisely in the WB.

We apologize for the confusion caused by the data presentation. The MET plasmid used to generate the isogenic clones contain a turbo-GFP tag which was described in the Methods (Pg 19, line 17). However, further details were not provided and clarified in the main text, which we have rectified throughout the manuscript. Indeed, the lower band (~140kDa) is the endogenous MET, whereas the top band (~170kDa) is the MET-tGFP exogenous protein (therefore not detected in the EV control. In the revised manuscript, we have named them (MET^{wt-tGFP} and MET^{N375S-tGFP}). The nomenclature of MET^{wt} and MET^{N375S} are now referring to the respective c-MET isoforms.

We also agree with the observation that the isogenic MET^{wt-tGFP} cells expressed relatively higher MET-tGFP than MET^{N375S-tGFP}, however we do not think that this could influence the conclusion drawn as a lower expression of MET-N375S drive a stronger oncogenic signal than MET-WT.

5. The authors showed increased cell migration, invasion, colony formation and metastases formation upon METN375S expression. None of these parameters have been addressed in Fig. S3. Therefore, the authors have not shown, that MET inhibition is not affective. Please provide data on cell migration, invasion, colony formation or metastases formation after MET inhibition.

We thank the reviewer for this suggestion. We will like to mention that the effect of MET inhibition (crizotinib) on invasion (Figure 6A) and 3D colony growth (Figure 6H) had been included. Nonetheless, we have compiled them together with the effect on cell migration in Figure S3. In summary, 1 μ M of crizotinib that is sufficient to abrogate p-MET is unable to prevent cellular migration, invasion and colony formation of MET^{N375S-tGFP} cells. These data have been discussed in the Result section (Pg. 7, line 16-18).

6. PLA data are not convincing: PLA signals seem to be localized in the perinuclear region, indicating ER localization. MET and HER2 should be preferentially located at the plasma membrane. Therefore please provide additional confocal analysis of METN375S and HER2 (co-)localization. So far the data presented only show interaction of HER2 and METN375S, however it remains unclear, if this is an indirect or direct interaction. Therefore, statements such as “METN375S to heterodimerize with HER2” (Abstract) should be toned down or additional data should be provided, validating a direct binding.

We are grateful to the reviewer for this suggestion, and we agree that performing confocal imaging will strengthen the case of MET^{N375S} and HER2 co-localization. Despite the seemingly perinuclear co-localization of MET/HER2, we think that this could be the limitation of wide-field immunofluorescence imaging where out-of-focus signals are captured. Our confocal analysis confirms the co-localization of MET^{N375S} and HER2 at the plasma membrane (Figure 4F), (Pg 8, line 20-22).

In addition, we have toned down the emphasis on HER2-MET^{N375S} heterodimerization in the Abstract to “MET^{N375S} to interact with HER2”. Also, we have included a statement “Collectively, the co-localization and strong binding affinity of both receptors in our assay models indicated that HER2 is a preferred interacting partner of MET^{N375S}.” on Pg 9, line 1-2. To provide additional data to support MET^{N375S} and HER2 binding, we have further conducted structure simulation to investigate the impact of a single amino acid substitution on the Sema domain. The simulation predicts for “significant localized conformational changes with the asparagine-to-serine substitution (Figure S14A,B), which likely expand the interacting surface of MET^{N375S} with HER2 (Figure S14C,D).” This preliminary data have

been included as Supplementary Figure 14, and discussed in the Discussion section (Pg 15, line 15-19).

Minor concerns:

1. Page 4, lane 26: How was amplicon-enriched NGS performed? There is no protocol in the manuscript and not in the given reference 19.

We apologize for the lack of explanation for this part, and we have provided the protocol in the Supplementary Methods. The amplicon-enriched protocol has been published in the supplementary information in reference 19 (Kong et al, Mol Cancer Ther 14, 1750-1760, 2015) as such:

“Extraction of gDNA from formalin-fixed, paraffin-embedded (FFPE) lung SCC tissues was carried out using QIAamp® DNA FFPE Tissue Kit (Qiagen). Library construction was performed with targeted exome enrichment using GeneRead DNAseq Gene Human Lung Panel (Qiagen) according to manufacturer’s protocol. Each set of enriched library was subjected to size-selection using GeneRead Size Selection kit (Qiagen) and AMPure Beads (Beckman Coulter). Multiplexing was conducted for barcoding of each library during DNA amplification. Exome sequencing on tumor DNA was performed with Illumina HiSeq2000 Platform. The mean total number of sequencing reads was 11,447,300, with 81% of target bases above 30x coverage. The generated sequencing reads were subjected to variant calling by Genome Analysis Toolkit (GATK) from Broad Institute. Significantly mutated genes ($P < 0.01$) were identified by mapping to public database (dbSNP) with a mean of 194 single nucleotide polymorphisms (SNPs) and 28 genomic rearrangements (insertions/deletions) per tumor. These genetic variants were subjected to functional annotation with the COSMIC Database for detection of cancer-related mutations.”

2. Fig. 3E: pMET and pHER2 IHC are depicted, but the text indicates pSrc instead of pHER2 (page 6, lane 5). Please clarify.

We apologize for the lack of clarity of this statement. The upregulation of p-Src was shown in Figure 3D (Western blotting) whereas the expression of p-HER2 was demonstrated in Figure 3D-E (Western blotting and IHC). We have rephrased the sentence to “In concordance, higher expression of phosphorylated Src (Fig. 3D) and MET (Fig. 3D,E) were detected in MET^{N375S} tumors.” (Pg 7, line 5)

3. Page 7, lane 2: The authors claim a hyperactive signaling of METN375S. Since only Src phosphorylation seems to be increased and MET inhibition has no significant influence, there is no obvious hyperactivity. Please use a moderate wording.

We have rephrase the wording to “oncogenic signaling”

4. There are no WB to control the knockdown shown in Fig. 2B, F, D. Please provide. The knockdown efficiency of MET have been provided as part of Figure 2.

5. Since there is no detailed analysis of nuclear MET, there is no real benefit from Fig. 4A, please shift it to supplementary information. Furthermore, the authors have shown, that crizotinib has no effect in terms of proliferation and tumor growth. Therefore I cannot follow the conclusions drawn from this experiments (page 7, lane 8); crizotinib likely does not induce stress to the cells.

After discussing with the co-authors, we agree with the reviewer that Figure 4A does not provide additional value to the manuscript. We have removed Figure 4A together with the confusing statement on cellular stress in relation to crizotinib treatment.

7. The observation, that crizotinib interrupts interaction with HER but does not influence the tumor growth or cell invasion might argue for additional players or interaction partners, who might be responsible for ongoing HER2 activity. Please discuss.

We agree with the reviewer that our data collectively suggest the involvement of additional players, which sustain HER2 activity despite MET inhibition. Given the pleiotropic nature of HER2, it is highly possible that the activated HER2 will form heterodimer with another member of the EGFR family, but this remains unclear at this point.

We have discussed this in several parts of the manuscripts: 1) once activated by heterodimerization, HER2 remains constitutively active despite kinase inhibition of its partner receptor (Pg 10, line 23-24), 2) Collectively, these data suggest that MET^{N375S-IGFP} cells attain the aggressive phenotype through intact MET and HER2 receptors, leading to HER2 phosphorylation that once activated is constitutively active and is irrepressible by MET kinase inhibition (Pg 11, line 9-12), 3) Our findings strongly indicate that phosphorylated MET^{N375S} leads to constitutively active HER2 that mediates signaling regardless of MET inhibition (Pg 16, line 1-3), 4) This insensitivity to MET inhibition is surprising and remains unexplained; possible reasons could include involvement of other HER2 activating mechanisms inhibition (Pg 16, line 3-5).

Reviewer #3 (Remarks to the Author): Expert in kinase biochemistry and structure

This manuscript by Goh and colleagues describes a common germline polymorphism of Met, N375S, that confers a more aggressive phenotype and poorer prognosis to squamous cell carcinomas, apparently through heterodimerization of the variant receptor with the Her2/Neu kinase. In cell lines and in xenografts, Her2 inhibitors (both antibodies and small molecules) are effective in inhibiting growth and downstream signaling of cells expressing MetN375S, whereas Met inhibitors have little if any effect in these cells. This effect can only be demonstrated for SCC (head and neck, lung), and not other tumors expressing MetN375S. Pilot clinical data for two patients with refractory MetN375S tumors treated with Her2 inhibitors are encouraging.

This is interesting a potentially quite important work, in identifying a novel therapeutic approach to treat tumors with a relatively common Met polymorphism. The mechanistic implications are also quite interesting, in that activating heterodimerization of receptor tyrosine kinases from different families is not a well-established activation mechanism. Unfortunately the precise mechanism of activation remains a bit mysterious, as Her2 activation seems to be maintained and even enhanced by Met inhibitors in MetN375S cells; whether a kinase-inactive N375S mutant (as opposed to a kinase domain deletion mutant) could still promote Her2 activation would be informative in this regard. However despite a number of remaining questions regarding mechanism, this work provides important new insights and sets the stage for future studies to explore clinical efficacy and molecular mechanism.

We are glad that the reviewer found the study to be interesting, and we appreciate the favorable remarks.

Specific points:

1. I found myself confused regarding the various cell models used for particular experiments.

Some use ectopic overexpression, and some specific knock-in of the N375S variant into the endogenous locus, and in some cases apparently a GFP fusion is used, but it is not always clear what cells were being used for particular experiments, and also whether the knock-in was homozygous or heterozygous for the variant allele.

We apologize for the confusions caused by the naming of the various cell lines. As MET^{N375S} polymorphism is more prevalent in East Asian population, we are unable to obtain lung and head and neck SCC cells harboring this mutation from any global biosource centres (including ATCC). Apart from the patient-derived cell lines (NPC7) established by our co-authors (Daniel Tan and Gopal Iyer), all cell lines reported in the study are either isogenic (H2170, Calu-1 clones) or CRISPR-cas9 knock-in. In addition, MET plasmid used to generate the isogenic clones contain a turbo-GFP tag which was described in the Methods (Pg 19, line 17). However, these details were missing in the initial submission, which we have rectified throughout the manuscript. For isogenic cell lines overexpressing MET, we have named them (MET^{wt-tGFP} and MET^{N375S-tGFP}), whereas homozygous MET knock-in is now labelled as MET^{N375S/N375S}. The nomenclature of MET^{wt} and MET^{N375S} are now referring to the respective c-MET isoforms. We hope that this will increase the clarity and improve the description of the manuscript.

2. For Fig. 3A and B, some more explanation is needed (e.g. how is “fold regulation” calculated; is this in fact a receptor tyrosine kinase antibody array, or more likely a phosphoprotein antibody array, etc.)

We apologize for our lack of clarity in the description of this figure. The assay performed was Proteome Profiler Human Phospho-kinase Array Kit that is a membrane-based antibody array for the determination of the relative phosphoprotein levels of several protein kinases. The labelling of “phospho-kinase array” and “RTK antibody array” was intertwined in our first draft, and it has been made consistent as Human phospho-kinase antibody array in the revised manuscript.

Detailed descriptions of the data analyses are now provided in the figure caption of Figure 3A and 3B, as well as in the Methods section (Pg 22, line 32 – Pg 23, line 3). Fold change for each target spots was analyzed in relative to MET^{wt-tGFP} cells, and presented as fold regulation (Positive fold regulation indicates relative fold increase in MET^{N375S-tGFP} cells; negative fold regulation indicates relative fold increase in MET^{wt-tGFP} cells).

3. On p. 12 and p. 14, the authors state that MetN375S phosphorylation is necessary for Her2 activation, but this seems in direct contradiction to their data with Met inhibitors, e.g. Fig. 5D, where Met phosphorylation is low yet Her2 phosphorylation is high. Further, there appears to be no heterodimerization of Met and Her2 in the presence of Met inhibitors. While I realize all the mechanistic details may not be worked out, some plausible mechanism(s) should be briefly discussed. Much is known from structural studies about the mechanisms of EGFR family activation via dimerization, which may be relevant here.

We agree with the reviewer that some of the mechanisms on this polymorphic MET remains unclear and paradoxical. Mainly, MET^{N375S} is critical for the activation (and recruitment) of HER2, but kinase inactivation of MET is unable to rescue this observation. We postulate that “upon its activation by MET^{N375S}, HER2 remains hyper-phosphorylated to continually promulgate oncogenic growth signals, further indicating that once activated by heterodimerization, HER2 remains constitutively active despite kinase inhibition of its partner receptor.” (Pg 10, line 21-24). We believe that this could be related to the complex mechanism of activation of HER2, whereby constitutive kinase activity could be achieved through receptor self-associated in a ligand-independent, concentration-dependent (such as in the case of HER2 amplification) manner.

Reviewer #4 (Remarks to the Author): Expert in kinase biochemistry and structure

The paper from Kong and colleagues is an impressive piece of work that describes a novel and unexpected alternative signaling mechanisms of a polymorphism (N275S) in the MET receptor tyrosine kinase in patients with certain head/neck and lung carcinomas. Based on solid epidemiological and clinical evidence on inferior outcome of patients with MET N275S, the authors use cellular models and mouse xenografts to show higher migration, transformation and metastatic potential of MET N275S. Biochemical analysis shows plausible downstream signaling differences when compared to MET wt. Careful pharmacological perturbation studies show low sensitivity for several MET TKIs in MET N275S cells and lead to the hypothesis that heterodimerization with HER2 may happen. Several well-controlled and state-of-the-art methods, including differential quantitative proteomics, PLA assays and amply controlled co-IPs in different cell lines unequivocally show HER2-MET N275S heterodimerization. Based on these findings, perturbation of MET N275S signaling with HER2 TKIs and antibodies results in inhibition of cell growth and signaling in cellular models in vitro, as well as in inhibition of tumor growth and prolongation of survival in mouse xenograft models. Finally, clinical efficacy of the proposed novel treatment approach is demonstrated in two patients. Overall, this is an exceptional paper in terms of amount and quality of data, clarity of presentation and how rigorous, systematic and well controlled the experiments were done.

We will like to thank the reviewer for the encouraging message.

(Very) minor points:

1. I was wondering about the relatively high (from 1-20 microM) GC50 values of the MET inhibitors in Figure S3. Do the authors think that the weak inhibitory activity is on-target? Given that authors describe CRISPR-Cas9 editing of these cell lines to make knock-ins (Fig. S2, page 5), have they also done CRISPR knock-outs for MET? If such MET knock-out cell lines would still show low sensitivity to MET TKIs, this would indicate off-target activity of the drugs (and further strengthen the data). Please comment.

We thank the Reviewer 4 for raising this interesting point of view. We are aware that the IC50 of MET inhibitors used in the study have all been previously reported as MET inhibitors, of which cabozantinib (VEGFR2, ROS1, MET) and crizotinib (ALK, MET) are well-characterized multi-kinase inhibitors. Tepotinib, on the other hand, is known to be a type Ib ATP-competitive selective MET inhibitor with no report on its off-target efficacy thus far. Among the tested compounds, only tivantinib has been reported to exhibit anti-tumor effect independent of MET (Katayama R, et al, 2013, Cancer Res; 73(10); 3087–96; Basilico C, et al, 2013 Clin Cancer Res. 1;19(15):4291; Calles A, et al, 2015 Mol Oncol. 9(1):260-9). In view of the vast amount of work conducted on these inhibitors (cabozantinib, tepotinib, crizotinib), we believe that the weak activity is on –target (as demonstrated by the strong inhibitory effect on MET phosphorylation at lower doses, Figure S3I). We have yet to conduct CRISPR-cas9 Knock-out of MET in our study. We will also like to point out that our reported IC50 of crizotinib are consistent with other reports in non-MET addicted cell lines (Katayama R, et al, 2013, Cancer Res; 73(10); 3087–96).

2. Some typos to correct:

page 14, line 22: "kinase", not "kinas"

This typo has been amended in the revised manuscript (Pg 15, line 22).

page 15, line 3: I think one word is missing. Either "leads to constitutively active HER2" or "leads to constitutive HER2 signaling".

This sentence has been amended in the revised manuscript to “leads to constitutively active HER2” (Pg 16, line 2).

page 15, line 30: "It is of note" not "In is of note"

This typo has been amended in the revised manuscript (Pg 16, line 30).

REVIEWERS' COMMENTS:

Reviewer #1 (Remarks to the Author):

This is a revised version of a study that unravels the mechanism of action behind aggressive squamous cell carcinomas. The finding couples a potential biomarker METN375S to a therapeutic intervention via HER2 inhibitors for this specific patient population.

The article covers a large amount of data that has been restructured in comparison to the previous version of the manuscript to create a more logical flow. More details were incorporated in the material and methods and results section for clarification.

The authors have addressed all comments and provided confirmatory data to address concerns of the reviewer. They have been provided either as revised figures or as Reviewer Figures. The data set that is present in Figure 2 of the revised manuscript supports the hypothesis of metastasis.

Addition of the molecular modeling of Figure S14 provides an orthogonal way to explain the enhanced affinity of the METN375S variant to HER2.

Publication of the revised version is recommended.

Reviewer #2 (Remarks to the Author):

For: Nature Communications

Kong et al. "A common MET polymorphism harnesses HER2 signaling to drive aggressive squamous cell carcinoma"

Ref: NCOMMS-19-15859-T

Corresponding Author: Boon Cher Goh

Reviewer #2 (Remarks to the Author): Expert in HNSCC

The authors addressed all concerns raised and improved the manuscript significantly. However, there are still some concerns to be addressed.

Concerns

Original comment

1. The authors state, that METN375S expression is relevant for SCC of the lung and the head and neck. For isogenic experiments only lung SCC cell lines were used but finally HNSCC patients were treated. Please provide some key experiments also for isogenic HNSCC cell lines.

Answer authors

Regarding the addition of cell lines, we have further generated isogenic HNSCC cells expressing either METwt-tGFP or METN375S-tGFP (UMSCC1 and SCC13). Key experiments were conducted to validate the GOF of the isogenic clones (invasion, 3D colony growth) and presented as Supplementary Figure 2. These observations were consistent with the observations in the patient-derived homozygous METN375S cell line (NCC-NPC7) (Supplementary Figure S9). These key findings demonstrate that METN375S expression could induce invasiveness and malignant transformation in both lung and head and neck SCC cells. Currently, our clinical trial studying HER2 inhibition in METN375S positive patients is open for recruitment of both lung and head and neck SCC patients. (ClinicalTrials.gov Identifier: NCT03938012)

Renewed demand

MET-WT-GFP and MET-N375S-GFP expression and phosphorylation can only hardly or even not be detected in the Western blots for SCC13 (here, there are two weak upper bands for N375S) and UMSCC1 of Fig. S2. There seems to be detection only of the endogenous MET. Have these cells also been transfected with the GFP-constructs and why is there no signal?

Original comment

6. PLA data are not convincing: PLA signals seem to be localized in the perinuclear region, indicating ER localization. MET and HER2 should be preferentially located at the plasma membrane. Therefore please provide additional confocal analysis of METN375S and HER2 (co-)localization. So far the data presented only show interaction of HER2 and METN375S, however it remains unclear, if this is an indirect or direct interaction. Therefore, statements such as "METN375S to heterodimerize with HER2" (Abstract) should be toned down or additional data should be provided, validating a direct binding.

Answer authors

We are grateful to the reviewer for this suggestion, and we agree that performing confocal imaging will strengthen the case of METN375S and HER2 co-localization. Despite the seemingly perinuclear co-localization of MET/HER2, we think that this could be the limitation of wide-field immunofluorescence imaging where out-of-focus signals are captured. Our confocal analysis confirms the co-localization of METN375S and HER2 at the plasma membrane (Figure 4F), (Pg 8, line 20-22).

In addition, we have toned down the emphasis on HER2-METN375S heterodimerization in the Abstract to "METN375S to interact with HER2". Also, we have included a statement "Collectively, the co-localization and strong binding affinity of both receptors in our assay models indicated that HER2 is a preferred interacting partner of METN375S." on Pg 9, line 1-2. To provide additional data to support METN375S and HER2 binding, we have further conducted structure simulation to investigate the impact of a single amino acid substitution on the Sema domain. The simulation predicts for "significant localized conformational changes with the asparagine-to-serine substitution (Figure S14A,B), which likely expand the interacting surface of METN375S with HER2 (Figure S14C,D)." This preliminary data have been included as Supplementary Figure 14, and discussed in the Discussion section (Pg 15, line 15-19).

Renewed demand

The confocal images showing MET and Her2 co-localization in Figure 4 F are not convincing. MET seems to be preferentially localized at in the cytosol. Please show the green and red channels also separately and single Z-levels (it seems to be a merged Z-stack). For wt MET, there seems to be co-localization with Her2 (yellow staining) as well, especially in the cytosol. Plotting the intensity distribution for each channel could help clarify these results.

Reviewer #3 (Remarks to the Author):

The authors have adequately addressed most of my concerns with the initial submission. The revised manuscript is substantially improved, and therefore I recommend publication.

One very minor suggestion: On p. 12, the NCC-NPC7 cell line is described as "HER2-". To me this nomenclature suggests lack of HER2 expression or function, which of course is not the case. To avoid confusion I suggest different description/terminology.

Reviewer #4 (Remarks to the Author):

The authors have added experiments and revised the manuscript. The points raised by myself after the initial submission were addressed. No further comments.

REVIEWERS' COMMENTS:

Reviewer #1 (Remarks to the Author):

This is a revised version of a study that unravels the mechanism of action behind aggressive squamous cell carcinomas. The finding couples a potential biomarker METN375S to a therapeutic intervention via HER2 inhibitors for this specific patient population.

The article covers a large amount of data that has been restructured in comparison to the previous version of the manuscript to create a more logical flow. More details were incorporated in the material and methods and results section for clarification.

The authors have addressed all comments and provided confirmatory data to address concerns of the reviewer. They have been provided either as revised figures or as Reviewer Figures. The data set that is present in Figure 2 of the revised manuscript supports the hypothesis of metastasis. Addition of the molecular modeling of Figure S14 provides an orthogonal way to explain the enhanced affinity of the METN375S variant to HER2. Publication of the revised version is recommended.

We thank the Reviewer for the recommendation.

Reviewer #2 (Remarks to the Author):

For: Nature Communications

Kong et al. "A common MET polymorphism harnesses HER2 signaling to drive aggressive squamous cell carcinoma"

Ref: NCOMMS-19-15859-T

Corresponding Author: Boon Cher Goh

Reviewer #2 (Remarks to the Author): Expert in HNSCC

The authors addressed all concerns raised and improved the manuscript significantly. However, there are still some concerns to be addressed.

Original comment

1. The authors state, that METN375S expression is relevant for SCC of the lung and the head and neck. For isogenic experiments only lung SCC cell lines were used but finally HNSCC patients were treated. Please provide some key experiments also for isogenic HNSCC cell lines.

Answer authors

Regarding the addition of cell lines, we have further generated isogenic HNSCC cells expressing either METwt-tGFP or METN375S-tGFP (UMSCC1 and SCC13). Key experiments were conducted to validate the GOF of the isogenic clones (invasion, 3D colony growth) and presented as Supplementary Figure 2. These observations were consistent with the observations in the patient-derived homozygous METN375S cell line (NCC-NPC7) (Supplementary Figure S9). These key findings demonstrate that METN375S expression could induce invasiveness and malignant transformation in both lung and head and neck SCC cells. Currently, our clinical trial studying HER2 inhibition in METN375S positive patients is open for recruitment of both lung and head and neck SCC patients. ([ClinicalTrials.gov](https://clinicaltrials.gov/ct2/show/study/NCT03938012) identifier: NCT03938012)

Renewed demand

MET-WT-GFP and MET-N375S-GFP expression and phosphorylation can only hardly or even not be detected in the Western blots for SCC13 (here, there are two weak upper bands for N375S) and UMSCC1 of Fig. S2. There seems to be detection only of the endogenous MET. Have these cells also been transfected with the GFP-constructs and why is there no signal?

We agree with the reviewer on this. It is indeed true that the expressions of MET-tGFP are weak in both HNSCC cells. Despite using the same plasmids with strong CMV promoter and the same transfection protocol, we noticed that both UMSCC1 and SCC13 have lower transfection efficiency as compared to the two LUSC cell lines (H2170 and Calu-1), as suggested by the lower GFP expression during selection process. One possible explanation is that both cell lines have been cultured for a long time, and therefore uptake efficiency of the transfected plasmid is lower. At this juncture, we are unable to obtain the cell lines at lower passage number, and we believe that this technical issue has significantly affected the MET expression of the stable clones. Given that the transfection efficiency is cell context dependent, we are therefore unable to control the expression of MET-tGFP protein consistently in all cell types, which is a limitation in this cellular model. Nonetheless, we would like to point out that the N375S variant exerts similar phenotypes in HNSCC that is consistent with our observations in the two LUSC cell lines as well as the CRISPR-edited clones.

We have now rerun the lysates and present a clearer blot on the presence of MET-tGFP proteins in UMSCC1 cells, albeit at a lower expression levels as compared to the endogenous MET (Figure xx). We conceded that the expression of MET-tGFP in UMSCC1 is extremely weak, particularly in UMSCC1 cells where p-MET is absent.

We would also like to point out that the weak upper band observed in these cell types could be the pro-MET isoform, as indicated by the manufacturer:

Reviewer Figure 1: Illustration of MET protein molecular weight by Cell Signaling Technology (<https://www.cellsignal.com/products/primary-antibodies/phospho-met-tyr1349-130h2-rabbit-mab/3133>)

Original comment

6. PLA data are not convincing: PLA signals seem to be localized in the perinuclear region, indicating ER localization. MET and HER2 should be preferentially located at the plasma membrane. Therefore please provide additional confocal analysis of METN375 and HER2 (co-)localization. So far the data presented only show interaction of HER2 and METN375, however it remains unclear, if this is an indirect or direct interaction. Therefore, statements such as “METN375S to heterodimerize with HER2” (Abstract) should be toned down or additional data should be provided, validating a direct binding.

Answer authors

We are grateful to the reviewer for this suggestion, and we agree that performing confocal imaging will strengthen the case of METN375S and HER2 co-localization. Despite the seemingly perinuclear co-localization of MET/HER2, we think that this could be the limitation of wide-field immunofluorescence imaging where out-of-focus signals are captured. Our confocal analysis confirms the co-localization of METN375S and HER2 at the plasma membrane (Figure 4F), (Pg 8, line 20-22). In addition, we have toned down the emphasis on HER2-METN375S heterodimerization in the abstract to “METN375S to interact with HER2”. Also, we have included a statement “Collectively, the co-localization and strong binding affinity of both receptors in our assay models indicated that HER2 is a preferred interacting partner of METN375S.” on Pg 9, line 1-2. To provide additional data to support METN375S and HER2 binding, we have further conducted structure simulation to investigate the impact of a single amino acid substitution on the Sema domain. The simulation predicts for “significant localized conformational changes with the asparagine-to-serine substitution (Figure S14A,B), which likely expand the interacting surface of METN375S with HER2 (Figure S14C,D).” This preliminary data have been included as Supplementary Figure 14, and discussed in the Discussion section (Pg 15, line 15-19).

Renewed demand

The confocal images showing MET and Her2 co-localization in Figure 4 F are not convincing. MET seems to be preferentially localized at in the cytosol. Please show the green and red channels also separately and single Z-levels (it seems to be a merged Z-stack). For wt MET, there seems to be co-localization with Her2 (yellow staining) as well, especially in the cytosol. Plotting the intensity distribution for each channel could help clarify these results. We thank the reviewer for this suggestion. We have provided individual green (MET) and red (HER2) channels for the confocal data. We would also like to clarify that the images shown are indeed single Z-panel, imaged using LSM800 at 40x magnification (Figure 4F) The authors would also like to point out that while both MET and HER2 are receptor tyrosine kinases, MET is known to be localized in both plasma membrane and cytosol (Protein Atlas database) while HER2 is predominantly a membrane protein. In order to increase the confidence of this finding, we have conducted additional analyses. Briefly, HER2 fluorescence intensity for each image is segmented and overlaid on to the MET channel (Reviewer Figure 2). The MET intensity that overlapped with the HER2 overlay is then quantified. This analysis aims to provide a measurement of the fraction of the total MET that overlapped (co-localized) with HER2 in each image. This analysis has been supplemented in Figure 4F.

In addition, the Reviewer also pointed out an observation made by Reviewer 1 earlier, that MET^{wt} interacts/co-localizes with HER2. This is likely due to the inherent binding affinity of MET to HER2, as reported by Yang L, et al. (Sci. Transl. Med. 11, eaav1620 (2019)), which could be reinforced under transient overexpression of MET. However, N375S exhibits way

stronger interaction to HER2 in our stable MET-tGFP expressing cells, suggesting that N375S is a stronger binding partner of HER2 with higher affinity compared to its WT counterpart. This is again reiterated by our structural simulation (Figure S12) that N375S mutation modified the Sema domain of MET to expose more interacting surface with HER2. This has been discussed in Pg 14, line 18-20.

Reviewer Figure 2: Data analyses on MET-HER2 co-localization. Green channel indicates MET fluorescence intensity. Image on the right is the segmented region based on HER2 fluorescence, that has been overlaid onto the green channel (blue highlighted region). The green intensity within the overlapping region is interpreted as the MET fraction that co-localized with HER2. This image is part of the illustration of Figure.4F.

Reviewer #3 (Remarks to the Author):

The authors have adequately addressed most of my concerns with the initial submission. The revised manuscript is substantially improved, and therefore I recommend publication.

One very minor suggestion: On p. 12, the NCC-NPC7 cell line is described as "HER2-". To me this nomenclature suggests lack of HER2 expression or function, which of course is not the case. To avoid confusion I suggest different description/terminology.

We thank the Reviewer for the suggestion. The terminology of HER2- has been clarified as "HER2-non-amplified" in the revised manuscript.

Reviewer #4 (Remarks to the Author):

The authors have added experiments and revised the manuscript. The points raised by myself after the initial submission were addressed. No further comments.

We thank the Reviewer for the recommendation.